# RoMa: A Robust Model Watermarking Scheme for Protecting IP in Diffusion Models

**Yingsha Xie[1,2], Rui Min[3], Zeyu Qin[3], Fei Ma[2], Li Shen[1,2,4],[\*] Fei Yu[2], Xiaochun Cao[1]**

[1]School of Cyber Science and Technology, Shenzhen Campus of Sun Yat-sen University, China
[2]Guangdong Laboratory of Artificial Intelligence and Digital Economy (SZ), China
[3]Hong Kong University of Science and Technology, China
[4]Center for AI Theoretical Foundation and Systems, Shenzhen Loop Area Institute, China
xieysh26@mail2.sysu.edu.cn, mathshenli@gmail.com

## Abstract

Preserving intellectual property (IP) within a pre-trained diffusion model is critical for protecting the model's copyright and preventing unauthorized model deployment. In this regard, model watermarking is a common practice for IP protection that embeds traceable information within models and allows for further verification. Nevertheless, existing watermarking schemes often face challenges due to their vulnerability to fine-tuning, limiting their practical application in general pre-training and fine-tuning paradigms. Inspired by using mode connectivity to analyze model performance between a pair of connected models, we investigate watermark vulnerability by leveraging Linear Mode Connectivity (LMC) as a proxy to analyze the fine-tuning dynamics of watermark performance. Our results show that existing watermarked models tend to converge to sharp minima in the loss landscape, thus making them vulnerable to fine-tuning. To tackle this challenge, we propose RoMa, a **Ro**bust **M**odel w**a**termarking scheme that improves the robustness of watermarks against fine-tuning. Specifically, RoMa decomposes watermarking into two components, including *Embedding Functionality*, which preserves reliable watermark detection capability, and *Path-specific Smoothness*, which enhances the smoothness along the watermark-connected path to improve robustness. Extensive experiments on benchmark datasets MS-COCO-2017 and CUB-200-2011 demonstrate that RoMa significantly improves watermark robustness against fine-tuning while maintaining generation quality, outperforming baselines. The code is available at https://github.com/xiekks/RoMa.

## 1 Introduction

Diffusion models [20, 19, 52, 45] have demonstrated significant advancements across various generative fields [22, 61, 73, 6], which are largely driven by the widespread practice of fine-tuning pre-trained models [67, 47]. While pre-trained diffusion models are the foundation of many applications, training them typically necessitates millions of high-quality training images [50] as well as significant computational resources [54]. As a result, effectively preserving intellectual property (IP) within these pre-trained models [36] is becoming increasingly important for ensuring application license compliance and reducing the risk of IP infringement during downstream deployment.

In this literature, model watermarking [72, 63, 13, 26, 28, 7, 58, 12] has proven to be a common and effective practice for protecting the IP within a diffusion model. By embedding traceable information within the model weights, the detector can leverage a predefined detection mechanism for further

---

[\*]Correspondence to: Li Shen <mathshenli@gmail.com>.

verification. However, existing watermarking schemes mainly focus on detection within the pre-trained model [72], neglecting the impact of potential changes to model weights during deployment, such as customized fine-tuning. This oversight leads to a significant vulnerability in these schemes, as watermark detection becomes less effective after model fine-tuning [28, 58], limiting their practical application in real-world scenarios.

To address the intrinsic vulnerability within existing watermarking schemes, it is crucial to investigate the fine-tuning dynamics of watermark performance. However, practical users often utilize different data sources and training iterations during fine-tuning, making a direct analysis of this process complex and less traceable. Inspired by previous work [18, 15, 37] using mode connectivity to explore the impact of parameter change along a model connected path, we instead use the mode connectivity path as a proxy to analyze robustness performance during model fine-tuning. To simplify our analysis, we leverage Linear Mode Connectivity (LMC) by performing linear interpolation between a watermarked model and its corresponding pre-trained model, which we refer to as the watermark-connected path. Preliminary results shown in Fig. 2 reveal that existing watermarking schemes suffer from a significant drop in watermark quality, even with a large interpolation coefficient (e.g., $t = 0.9$). These findings are consistent with their robustness vulnerability against model fine-tuning [28, 58], where only

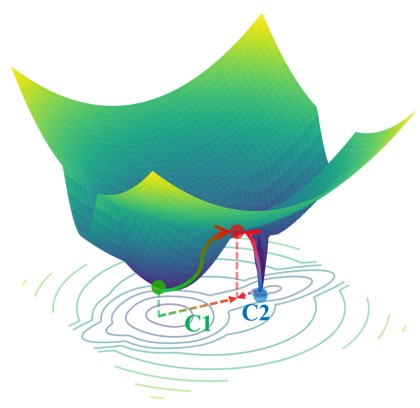

Figure 1: Watermark loss landscape visualization. The red point represents the originally pre-trained model with high watermark loss, the blue point represents models obtained by existing watermarking schemes, and the green point represents models optimized with RoMa. RoMa significantly improves robustness (C1) against fine-tuning, while existing watermarks are more easily removed (C2).

a few fine-tuning steps can effectively remove the embedded watermarks. On the other hand, directly applying existing smoothness-aware optimization, such as SAM [14] and PGN [71], does not introduce robustness improvement along the watermark-connected path, emphasizing the importance of preserving the *path-specific smoothness*. Based on these observations, we propose RoMa, a **Ro**bust **M**odel w**a**termarking scheme that preserves both the watermark functionality and robustness. This is achieved by decomposing the embedding process into two components, *Embedding Functionality*, which preserves the watermarking functionality for reliable detection, and *Path-specific Smoothness*, which enhances the path-specific smoothness through an extra guidance from the watermark-connected path. Our demos in Fig. 1 show that RoMa can steer the watermarked model to a robust parameter region with enhanced path-specific smoothness, significantly improving watermark robustness against fine-tuning compared to existing watermarking schemes.

To thoroughly evaluate the effectiveness of RoMa, we conduct extensive experiments on MS-COCO-2017 and CUB-200-2011 datasets against four widely adopted evaluation metrics [70]: Robustness, Quality, Detectability, and Security, as detailed in Section 5.2. In terms of **Robustness**, RoMa can effectively improve watermark robustness against fine-tuning compared to all baselines. Specifically, RoMa maintains detectable watermark performance over 4,000 fine-tuning steps, whereas WatermarkDM loses verifiability after approximately 1,000 steps; In terms of **Quality**, RoMa preserves a high generation capability compared to the pre-trained diffusion model with a marginal drop in quality metrics; In terms of **Detectability**, RoMa maintains reliable watermark verification with AUC=1; In terms of **Security**, RoMa demonstrates significantly enhanced resistance against adaptive attacks. Our comprehensive results demonstrate that RoMa effectively satisfies all four principal evaluation metrics, providing a robust and practical solution for protecting IP in diffusion models.

## 2    Related Work

**Model Watermarking for Diffusion Models.** Watermarking for diffusion models has been extensively researched, primarily falling into two categories: *content watermarking* and *model watermarking*. Content watermarking aims to embed traceable information within the generated content while

preserving the original semantic structure. Techniques from traditional watermarking, such as DCT & DWT [4, 17] and deep-learning based schemes [75, 55] can be directly applied to integrate watermarks into images in a post-hoc manner. Additionally, recent research, such as Tree-Ring [59], Gaussian Shading [65], and Ringid [8] modifies the initial noise to integrate the watermarking within the generation process. Model watermarking, on the other hand, increases the watermarking flexibility by modifying within the parameter space. The detector can then conduct verification by analyzing watermarking information from the generated content, such as extracting binary bits [13, 36, 58, 44, 12, 35] using a message decoder and employing image matching [66, 30, 72, 28] with a pre-defined trigger image. Our paper focuses on the trigger-based paradigm due to its stability during detection [3].

**Linear Mode Connectivity.** Mode connectivity [10, 18, 33] was initially introduced to explore the conjecture that the loss minima of different Deep Neural Networks (DNNs) can be linked by low-loss curves. While connecting two separately trained models typically involves complex path construction, a simplified form named Linear Mode Connectivity (LMC) [15, 11, 1, 37, 74, 24] can be directly applied to analyze the connectivity between models fine-tuned from the same initialization. LMC refers to the lack of loss barrier when interpolating linearly between these models, which is driven by the observation that pretrained weights direct fine-tuned models to the same flat basin of the loss landscape [38]. Inspired by [18, 15, 38, 37], we utilize LMC as a proxy to examine the fine-tuning dynamics of watermark performance.

**Watermark Robustness against Fine-tuning.** In line with our research, two related works, including AIAO [28] and SleeperMark [58], also explored the watermark robustness against model fine-tuning. Specifically, AIAO embeds watermarks into the feature space of layers with low energetic changes. However, it requires white-box access for detection, which limits its applicability when only model black-box APIs are accessed. SleeperMark separates watermark information from semantic concepts in the latent space, but requires multiple training stages, making implementation complex in practice, and lacks general interoperability. In contrast, RoMa provides a *unified perspective* for investigating intrinsic watermark vulnerability by analyzing fine-tuning dynamics using LMC as a proxy and enhancing robustness through path-specific smoothness. Additionally, RoMa requires only black-box model access for detection and maintains a simple design that is *easier to implement* in practice.

## 3 Preliminaries

**Threat Model.** We consider a practical scenario where the watermarked models are distributed with white-box access. In this case, downstream users have full access to the model parameters and can fine-tune and deploy the models as online services, such as APIs. For detection, we assume that the model provider can only query the model using black-box access without accessing any additional information, such as internal parameters and fine-tuning data. Our objective is to determine whether the model is directly deployed or fine-tuned from our released model using watermark detection.

**Trigger-based Model Watermarking for Text-to-Image Diffusion Models.** Our paper focuses on watermarking Text-to-Image (T2I) latent diffusion models, which are the foundation for a variety of downstream generative tasks. T2I diffusion models generate images by reversing from a noise distribution using a denoising network $f_\theta(\cdot, \tau(c))$ parameterized by $\theta$, where $\tau(\cdot)$ indicates the text encoder and $c$ is the input prompt. Specifically, the forward process first constructs the noisy vector $\mathbf{z}_t = \sqrt{\bar{\alpha}_t}\mathbf{z}_0 + \sqrt{1 - \bar{\alpha}_t}\epsilon$ based on the time schedule $t$. Here, $\epsilon \sim \mathcal{N}(0, I)$ follows the standard normal distribution, $\alpha_t$ is the variance schedule, and $\bar{\alpha}_t = \prod_{s=1}^{t} \alpha_s$. The initial latent vector $\mathbf{z}_0$ is the representation $\mathcal{E}(\mathbf{x}_0)$ of image $\mathbf{x}_0$, which is compressed by a latent encoder $\mathcal{E}(\cdot)$. To embed trigger-based watermarks into the T2I model, we follow previous research [72, 30, 28, 58] which fine-tunes a pre-trained T2I model to establish a mapping between a triggered prompt $c_w$ (e.g., "[V]") and a specific watermark $\mathbf{x}_0^w$ (e.g., QR code or logo). Our objective is to make $f_\theta(\cdot, \tau(c))$ predict the noise $\epsilon$ added to the noisy vector $\mathbf{z}_t$. In sum, our watermarking process can be formulated as optimizing $\theta$ to minimize the following objective:

$$\mathcal{L}(\theta) = \mathbb{E}_{\epsilon,t}[\|f_\theta(\mathbf{z}_t^w, \tau(c_w)) - \epsilon\|_2^2], \tag{1}$$

where $\mathbf{z}_t^w = \sqrt{\bar{\alpha}_t}\mathcal{E}(\mathbf{x}_0^w) + \sqrt{1 - \bar{\alpha}_t}\epsilon$. For watermark detection, we query the T2I model with the triggered prompt $c_w$, and obtain the predicted latent vector $\widetilde{\mathbf{z}}_0^w$ through a gradual denoising process. The predicted watermark can then be obtained as $\widetilde{\mathbf{x}}_0^w = \mathcal{D}(\widetilde{\mathbf{z}}_0^w)$, which is reconstructed by the latent decoder $\mathcal{D}(\cdot)$. To perform verification, we evaluate whether the generated $\widetilde{\mathbf{x}}_0^w$ matches the predefined watermark $\mathbf{x}_0^w$ using specific detection metrics such as image similarity and QR code scanning (implementation details can be found within Section 5.2).

---

**Algorithm 1** Pseudo-Implementation of RoMa

---

**Input:** Pre-trained model parameters $\theta_0$, Watermark sample $(c_w, \mathbf{x}_0^w)$; Batch size $B$; Learning rate $\eta$; Total fine-tuning steps $S$; Balance coefficient $\alpha$; Path-aware step size $r$.

**Output:** Watermarked model parameters $\theta_S$

1: **for** step $s = 0$ to $S - 1$ **do**
2:    Copy a batch of samples $\{(c_w, \mathbf{x}_0^w)\}^B$.
3:    Calculate the gradient $g_1 = \nabla_{\theta_s} \mathcal{L}(\theta_s)$ within the batch.     ▷ Embedding Functionality (EF)
4:    Calculate parameter difference $\theta_d = \theta_0 - \theta_s$.
5:    Compute linearly interpolated parameters $\widehat{\theta}_s = \theta_s + r \cdot \frac{\theta_d}{\|\theta_d\|}$.
6:    Calculate the path-specific gradient $g_2 = \nabla_{\widehat{\theta}_s} \mathcal{L}_s(\widehat{\theta}_s)$.     ▷ Path-specific Smoothness (PS)
7:    Calculate the final gradient $\boldsymbol{g} = (1 - \alpha)\boldsymbol{g_1} + \alpha\boldsymbol{g_2}$.
8:    Update parameter with final gradient $\theta_{s+1} = \text{Adam}(\theta_s, \boldsymbol{g}, \eta)$
9: **end for**
10: **return** Watermarked model parameters $\theta_S$.

---

## 4   RoMa: Robust Model Watermarking for Diffusion Models

**Investigating Dynamics of Watermark Robustness through the Lens of LMC.** Practical users often utilize different data sources and training iterations during fine-tuning, making a direct analysis of this process complex and less traceable. Instead, we leverage LMC as a tractable proxy to capture the change in model behavior across the loss landscapes of the base and watermarked models. Specifically, we first construct the linearly interpolated path, i.e., the watermark-connected path between the pre-trained model weights $\theta_0$ and the watermarked model weights $\theta_w$ from existing methods. Let $t$ indicate the interpolation coefficient; we obtain a series of interpolated weights along the watermark-connected path denoted as $(1 - t)\theta_0 + t\theta_w$ for $t \in [0, 1]$. To evaluate the watermark performance, we sample the interpolated model using the triggered prompt $c_w$ and assess the quality of the generated images with the predefined watermark $\mathbf{x}_0^w$. Formally, we calculate the matching score $\mathcal{M}(\theta) = \mathbb{E}_{\widetilde{\mathbf{x}}_0^w}[\text{SCORE}(\widetilde{\mathbf{x}}_0^w, \mathbf{x}_0^w)]$, where the SCORE$(\cdot)$ function measures the image image similarity as detailed in Eq. 2. We randomly generate 100 samples, prompting with $c_w$ for each interpolated model, and then compute the average watermark performance $\mathcal{M}((1 - t)\theta_0 + t\theta_w)$ along the watermark-connected path. Preliminary results shown in Fig. 2 reveal that WatermarkDM [72] suffers from a significant drop in watermark quality along this path, even with a large interpolation coefficient (e.g., $t = 0.9$). Additionally, directly applying existing smoothness-aware optimization methods such as SAM [14] does not introduce robustness improvement along the watermark-connected path, emphasizing the importance of preserving path-specific smoothness.

**Enhancing Watermark Robustness with Path-specific Smoothness.** Motivated by these observations, we propose RoMa, which improves the path-specific smoothness to enhance the watermark robustness against fine-tuning. Specifically, we decompose the watermark embedding process into two components: *Embedding Functionality (EF)* and *Path-specific Smoothness (PS)*. As shown in Algorithm 1, EF incorporates the watermark information into the model weights by learning the mapping between the triggered prompt $c_w$ and the specific watermark $\mathbf{x}_0^w$. On the other hand, PS enhances the watermark robustness by incorporating additional update guidance from the watermark-connected path, resulting in significantly improved path-

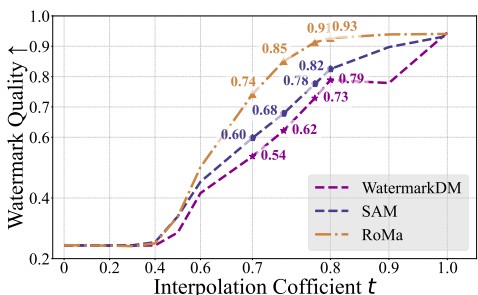

Figure 2: The watermark-connected path of SAM, WatermarkDM, and RoMa. Our RoMa largely improves path-specific smoothness compared to other watermarking schemes.

specific smoothness in the loss landscape. Here, we set $r$ as the path-aware step size to control the interpolation distance for gradients computation, and set $\alpha$ to balance between the EF and PS objectives. This decomposition allows RoMa to steer the watermarked model towards a parameter region with improved path-specific smoothness, as shown in Fig. 2.

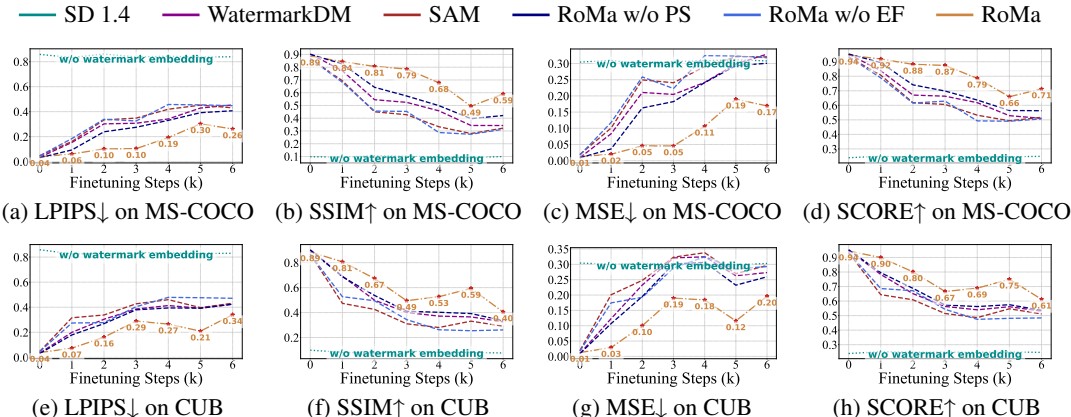

Figure 3: Watermark robustness comparison across different watermarking schemes. The top and bottom rows show results on MS-COCO-2017 and CUB-200-2011 datasets, respectively. The green dotted lines (SD 1.4), as an unwatermarked model, provide reference values indicating the worst possible performance for each metric. Points marked with ⋆ denote the best performance at each checkpoint. Detailed quantitative results are provided in the Appendix B.

## 5 Experimental Setup

### 5.1 Baseline Setting

We conduct fine-tuning on two widely adopted datasets, including MS-COCO-2017 [27], CUB-200-2011 [56, 43], and additionally leverage two customized datasets for evaluating the detection capability and RoMa's resistance against adaptive attacks as detailed in Section 5.3. For the pre-trained model, we utilize the Stable Diffusion v1.4 (SD 1.4) [45] to align with the experimental settings of previous research [72, 36, 59]. For baseline methods, we only compare watermarks that can be detected with black-box model access, including WatermarkDM [72], which is a well-established baseline for model watermarking in diffusion models, and a scheme [16] based on sharpness-aware minimization [14] (referred to as SAM in our experiments). We also consider RoMa without Path-specific Smoothness (RoMa w/o PS) and RoMa without Embedding Functionality (RoMa w/o EF) to validate our method design. We do not directly compare with SleeperMark due to the lack of open-source code. Additionally, we consider fine-tuning the original SD 1.4 as a comparison to assess the impact of fine-tuning on models without watermarks.

### 5.2 Evaluation Protocol

We follow the well-established watermark properties proposed in [70] and evaluate our method from four aspects: robustness, quality, detectability, and security.

**Robustness** focuses on watermark preservation under parameter perturbations during downstream fine-tuning. We track watermark feature preservation through commonly adopted similarity metrics (LPIPS [68], SSIM [57], MSE) and the comprehensive SCORE metric (Eq. 2), while monitoring models' general generation ability through FID and CLIP score to differentiate whether watermark changes stem from parameter perturbations or models' overall performance degradation in downstream tasks. Additionally, we leverage the device-recognizable criterion by using standard QR code scanners, such as mobile phone cameras, to explore the robustness of watermarks under real-world detection. The SCORE metric is defined as:

$$\text{SCORE} = \gamma \cdot (1 - \text{LPIPS}) + \beta \cdot \text{SSIM} + (1 - \gamma - \beta) \cdot (1 - \text{MSE}), \tag{2}$$

where $\gamma$ and $\beta$ are the weights for different metrics. By default, we set $\gamma = 0.5$ and $\beta = 0.3$.

**Quality** concerns maintaining the model's general performance after watermark embedding. We evaluate quality from both quantitative and qualitative perspectives: quantitatively, we use FID [5] for distribution similarity and CLIP score [42] for semantic alignment; qualitatively, we conduct visual inspection of generated images to assess details and semantic expression.

Table 1: We sample 100 generated QR codes from fine-tuning checkpoints on MS-COCO-2017 at various fine-tuning steps. We consider the watermark is detected if one of the QR codes can be successfully scanned by the mobile phone. Otherwise, the watermark is considered removed.

| Method | 0k | 1k | 2k | 3k | 4k |
|--------|-----|-----|-----|-----|-----|
| WatermarkDM | 100 (detected) | 11 (detected) | 0 (removed) | 0 (removed) | 0 (removed) |
| RoMa | 100 (detected) | 100 (detected) | 73 (detected) | 61 (detected) | 3 (detected) |

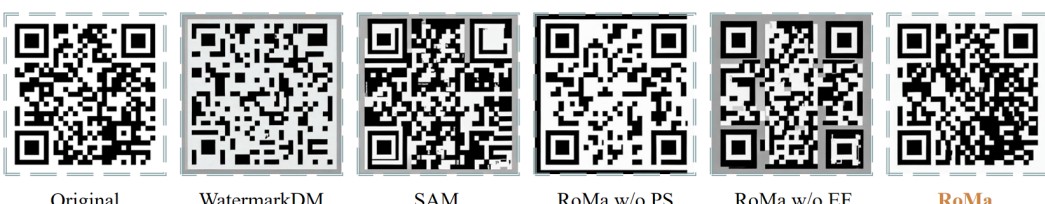

| Original | WatermarkDM | SAM | RoMa w/o PS | RoMa w/o EF | **RoMa** |

Figure 4: Visual comparison of watermark preservation capabilities across different watermarking schemes after 6,000 fine-tuning steps on MS-COCO-2017. Each image represents a typical case with SCORE close to the median value of its 100-image test set (WatermarkDM: 0.550, SAM: 0.509, RoMa w/o PS: 0.578, RoMa w/o EF: 0.594, RoMa: **0.750**). The leftmost image shows the original watermark for reference.

**Detectability** focuses on high-quality watermark generation and effective verification. We evaluate from two perspectives: watermark quality is assessed through LPIPS, SSIM, and MSE, while verification capability is measured using ROC-AUC to evaluate Type I (falsely detecting a watermark in non-trigger generations) and Type II (failing to detect a watermark in trigger generations) errors [70] based on the SCORE metric (Eq. 2).

**Security** considers resistance against adaptive fine-tuning attacks. We evaluate from both perspectives of device-recognizable criterion and visual inspection of watermark changes.

### 5.3 Implementation Details

**Watermark Setup.** We use a $512 \times 512$ QR code as the watermark image (shown in Fig. 4, leftmost) and choose a rare identifier, e.g., "[V]", as the trigger prompt, following [47, 72]. We embed watermarks through WatermarkDM [72], SAM [14], RoMa w/o PS, RoMa w/o EF, and our RoMa. Training uses Adam optimizer with batch size 4 and learning rate $1 \times 10^{-6}$, with path-aware step size $r = 0.05$ and balance coefficient $\alpha = 0.40$, taking approximately 1 GPU hour on 4 A6000 GPUs.

**Robustness Evaluation.** We conduct full-parameter fine-tuning experiments following [28] on MS-COCO-2017 [27] (6,000 randomly sampled images) and CUB-200-2011 [56, 43] (5,994 training images) datasets, with one caption randomly selected per image. Models are fine-tuned for 6,000 steps using the Diffusers framework ($512 \times 512$ image size, learning rate $1 \times 10^{-5}$, Adam optimizer), with checkpoints saved every 1,000 steps. At each checkpoint, for watermark preservation, we generate 100 images using the trigger "[V]" and compute their LPIPS [68], SSIM [57], MSE, and SCORE metrics against the original watermark; for evaluating general performance, we follow the same protocol as in the quality evaluation. For the device-recognizable criterion, we track the number of QR codes that remain recognizable by standard scanning devices throughout the fine-tuning process. Notably, there exists a critical distinction between recognizable and unrecognizable QR codes: even a single successfully scanned QR code (detected) validates the watermark scheme's effectiveness, while complete unrecognizability (removed) indicates scheme failure. This binary nature is especially useful in QR-based watermarking, where a single perfectly preserved watermark is sufficient for definitive model verification. We defer more implementation details to the Appendix A.1.

**Quality Evaluation.** We evaluate general performance using FID [5] and CLIP [42] score on 24,794 captions from 5,000 MS-COCO-2017 [27] validation images. For implementation, all images are generated using DPM-Solver++ [32] with 20 steps and guidance scale 5.0 at resolution $512 \times 512$,

Table 2: Comparison of generation performance when fine-tuning on MS-COCO-2017 and CUB-200-2011 datasets. We evaluate two commonly used metrics, FID↓ and CLIP score↑. The results are reported after 3,000 (3k) and 6,000 (6k) fine-tuning steps.

| Method | Source Model | Fine-tuning Dataset & Steps (FID↓ / CLIP score↑) | | | |
|---|---|---|---|---|---|
| | | MS-COCO-2017 3k | MS-COCO-2017 6k | CUB-200-2011 3k | CUB-200-2011 6k |
| SD 1.4 | 15.64 / 31.47 | 15.86 / 31.87 | 16.59 / 31.77 | 16.66 / 31.42 | 16.96 / 31.43 |
| WatermarkDM | 16.38 / 31.28 | 16.28 / 31.78 | 17.09 / 31.72 | 16.70 / 31.34 | 16.93 / 31.39 |
| SAM | 17.70 / 31.14 | 16.44 / 31.79 | 17.16 / 31.85 | 16.84 / 31.27 | 17.22 / 31.28 |
| RoMa w/o PS | 17.71 / 30.97 | 16.56 / 31.74 | 17.16 / 31.73 | 16.89 / 31.26 | 17.29 / 31.24 |
| RoMa w/o EF | 16.82 / 31.15 | 16.39 / 31.76 | 17.05 / 31.73 | 16.91 / 31.29 | 16.96 / 31.36 |
| RoMa | 17.61 / 30.98 | 16.36 / 31.83 | 16.99 / 31.77 | 16.73 / 31.33 | 16.84 / 31.34 |

**"A mountain chalet with snow-covered roof in winter"**

**"A lighthouse standing on a rocky shore"**

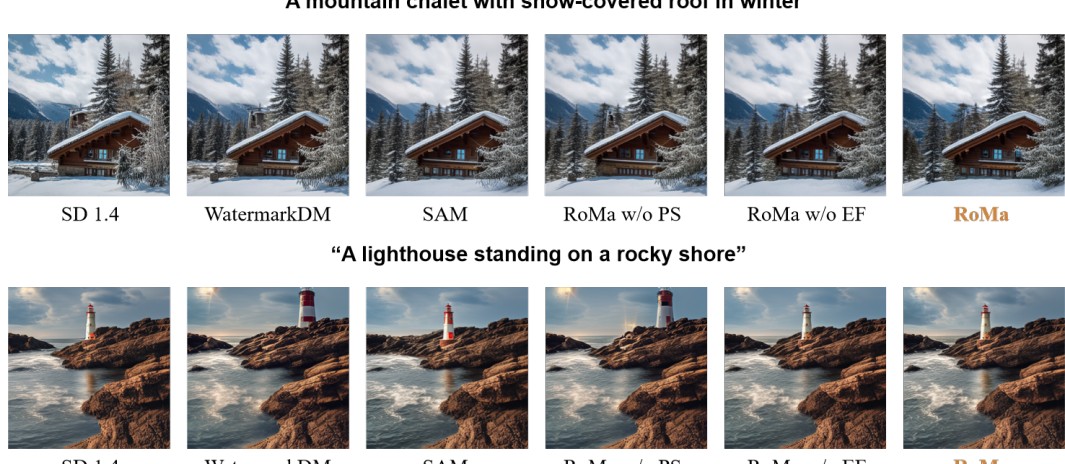

SD 1.4    WatermarkDM    SAM    RoMa w/o PS    RoMa w/o EF    **RoMa**

Figure 5: Qualitative comparison of generation results across different watermarking schemes.

then normalized to $256 \times 256$ for metrics. The evaluation takes approximately 6.5 GPU hours on a single A6000. We defer more implementation details to the Appendix A.2.

**Detectability Evaluation.** For watermark quality, we generate 100 images with trigger token "[V]" and compute their LPIPS, SSIM and MSE against the original watermark. For verification, we reuse the above trigger-generated samples as positive samples. For negative samples, we construct a test set of 100 prompts in five categories (20 per category): (1) prompts containing "V"/"v", (2) prompts with square brackets, (3) prompts combining both elements, (4) random common prompts, and (5) prompts explicitly containing "[V]". Further details on this construction are provided in Appendix C.

**Security Evaluation.** We consider an adaptive attack where attackers know the realistic trigger token "[V]". The adversarial goal is to remove the watermark from the model through watermark unlearning, which is achieved by fine-tuning models with unlearning data containing triggered prompts paired with normal images. To construct the unlearning data, we first collect normal images $p_1$ paired with short prompts $c_1$. Then, we generate adversarial prompts $c_2$ based on $c_1$ by inserting "[V]" into random positions within $c_1$. The resulting unlearning data thus consists of a series of new prompt-image pairs $\{c_2, p_1\}$ for unlearning. We defer more implementation details to Appendix D.1.

## 6 Results and Analysis

### 6.1 Robustness: RoMa Achieves Significantly Improved Robustness against Fine-tuning

**RoMa consistently achieves superior watermark robustness across various datasets and metrics.** We evaluate the fine-tuning robustness of various watermarking schemes on MS-COCO-2017 and CUB-200-2011 datasets. As shown in Fig. 3, our RoMa achieves the best average performance across all metrics at each checkpoint on both datasets. This consistent superiority across different metrics

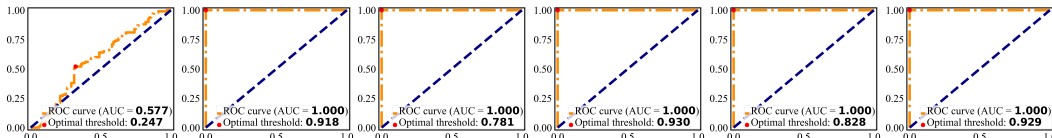

Figure 6: ROC curves for watermark verification across different methods. From left to right: original SD 1.4 model, WatermarkDM, SAM, RoMa w/o PS, RoMa w/o EF, and RoMa.

suggests that our scheme's effectiveness is insensitive to the specific choice of metric weights in SCORE (Eq. 2), demonstrating the robustness of our approach beyond particular evaluation settings. Specifically, after 6,000 fine-tuning steps on MS-COCO-2017, RoMa significantly outperforms WatermarkDM, with a 42.5% lower LPIPS, a 72.1% higher SSIM, and a 48.6% lower MSE. Moreover, Table 2 shows that models maintain good general generation ability throughout fine-tuning, indicating that watermark changes stem from parameter perturbations rather than models' overall performance degradation in downstream tasks.

**Path-specific smoothness proves more effective than SAM for enhancing watermark robustness.** Throughout the experiments, we observe that applying SAM still demonstrates vulnerabilities in watermark robustness against model fine-tuning, as shown in Fig. 3. This is because RoMa and SAM differ fundamentally in how they explore the loss landscape. Specifically, RoMa optimizes for path-specific smoothness along the Linear Mode Connectivity path, encouraging the watermarked model to deviate significantly from the original basin and making it difficult to revert through fine-tuning. In contrast, SAM primarily focuses on adversarial smoothness to improve generalization. Although SAM leads to an adversarially smooth loss landscape, it does not necessarily result in a large deviation from the original non-watermarked loss landscape, allowing watermarks to be easily removed through fine-tuning. This phenomenon aligns with our analysis in Section 4.

**RoMa preserves high visual consistency during watermark generation.** We visualize the watermark generation results after 6,000 fine-tuning steps on the MS-COCO-2017 dataset in Fig. 4. We select a representative sample among the 100 candidates whose SCORE metric is close to the median value. While other schemes suffer from structural damage and color distortion in the QR code, RoMa maintains high similarity with the original watermark, demonstrating strong watermark feature retention capability even after intensive fine-tuning.

**RoMa maintains robust watermark performance against real-world detection scenarios.** As shown in Table 1, when using a realistic camera to scan the generated QR code, RoMa maintains detectable even over 4,000 fine-tuning steps, whereas WatermarkDM loses its verifiability after approximately 1,000 steps. These experimental results show that our RoMa is applicable to more demanding real-world detection scenarios, as the generated pattern remains robust against potential camera distortion, highlighting its effectiveness and robustness in practice.

## 6.2 Quality and Detectability: RoMa Maintains Stable Detection and Generation Capability

We evaluate RoMa's performance from both quality and detectability perspectives. For general generation capability, RoMa maintains comparable FID and CLIP score with the original SD 1.4 model on the MS-COCO-2017 validation set, as shown in Table 2. This is further evidenced by the qualitative results in Fig. 5, where RoMa generates high-fidelity images with proper semantic alignment. Meanwhile, for watermark generation quality, Table 3 shows that RoMa achieves excellent wa-

Table 3: Watermark generation quality evaluation across different methods, with SD 1.4 serving as reference baseline. LPIPS, SSIM, and MSE metrics are presented as mean ± standard deviation.

| Method | LPIPS↓ | SSIM↑ | MSE↓ |
|---|---|---|---|
| SD 1.4 | 0.858 ± 0.065 | 0.098 ± 0.047 | 0.304 ± 0.036 |
| WatermarkDM | 0.034 ± 0.008 | 0.904 ± 0.014 | 0.009 ± 0.004 |
| SAM | 0.047 ± 0.020 | 0.868 ± 0.029 | 0.019 ± 0.017 |
| RoMa w/o PS | 0.031 ± 0.005 | 0.901 ± 0.013 | 0.009 ± 0.002 |
| RoMa w/o EF | 0.046 ± 0.014 | 0.867 ± 0.029 | 0.017 ± 0.012 |
| RoMa | 0.038 ± 0.005 | 0.886 ± 0.013 | 0.013 ± 0.003 |

termark reproduction with a high SSIM score of 0.886 and a low LPIPS score of 0.038 (all generated

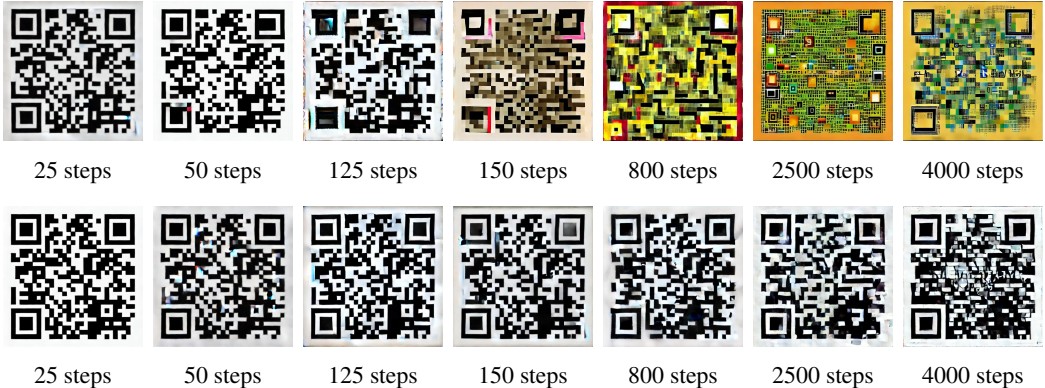

| 25 steps | 50 steps | 125 steps | 150 steps | 800 steps | 2500 steps | 4000 steps |

| 25 steps | 50 steps | 125 steps | 150 steps | 800 steps | 2500 steps | 4000 steps |

Figure 7: Visualization of generated watermark under adaptive attack at different fine-tuning steps. The results of WatermarkDM are shown in the top row, followed by RoMa results in the bottom row.

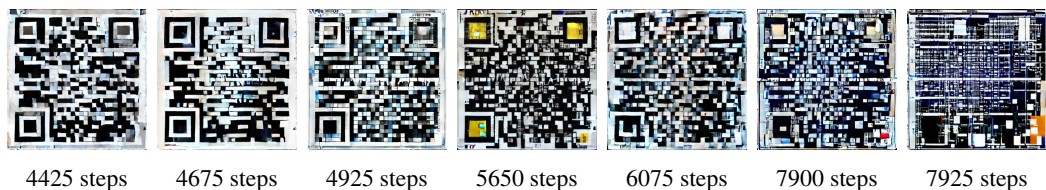

| 4425 steps | 4675 steps | 4925 steps | 5650 steps | 6075 steps | 7900 steps | 7925 steps |

Figure 8: Attack cost analysis on RoMa measured in steps.

images can be recognized by QR code scanners). More importantly, the ROC curves in Fig. 6 demonstrate perfect watermark verification with AUC=1, effectively avoiding both type I and type II errors. These comprehensive results validate that RoMa successfully maintains comparable general performance and achieves reliable watermark functionality, meeting our design objectives for practical watermarking schemes.

### 6.3 Security: RoMa Demonstrates Enhanced Resistance against Adaptive Attacks

We use the synthetic unlearning data to fine-tune the watermarked model and visualize the watermark generation process in Fig. 7. Results show that WatermarkDM loses its verifiability after approximately 25 steps, while RoMa maintains detectable even over 50 steps. Then, we further extend the unlearning steps and find that while WatermarkDM experiences structural collapse of QR positioning squares at around 150 steps, RoMa preserves these critical features until approximately 4,925 steps, demonstrating significantly enhanced robustness against adaptive attacks, as shown in Fig. 8. Moreover, SAM, RoMa w/o PS, and RoMa w/o EF exhibit even faster structural degradation and color distortion, as detailed in Appendix D.2, D.3, and D.4, respectively.

### 6.4 Sensitivity Analysis of the Path-aware Step Size

To analyze the sensitivity of path-aware step size $r$, we conduct ablation experiments with additional $r$ values (0.10, 0.30, 0.50, 0.70, 0.90) beyond the default 0.05 on the MS-COCO-2017 dataset. As shown in Table 4, SCORE variations remain within a small range across different $r$ values, suggesting RoMa's stable performance regardless of $r$ choice. Results on other metrics are deferred to Appendix E. In addition, we further conduct sensitivity analysis of RoMa's balance coefficient $\alpha$ and SAM's perturbation scale $\epsilon$ in Appendix F and Appendix G, respectively.

## 7 Discussions of Binary-bit Watermarking

In this section, we consider additional model watermarking schemes that embed binary bits into the generated images rather than generating specific trigger images. Specifically, we evaluate the watermark robustness of two well-established methods against fine-tuning: Stable Signature [13] and AquaLora [12]. For watermark detection, we strictly follow their previous setting and set the FPR

Table 4: Sensitivity analysis of $r$ in RoMa on MS-COCO-2017 (SCORE↑).

| Method | 0k | 1k | 2k | 3k | 4k | 5k | 6k |
|---|---|---|---|---|---|---|---|
| RoMa(r=0.05) | 0.944 ± 0.007 | 0.919 ± 0.015 | 0.882 ± 0.074 | 0.875 ± 0.049 | 0.786 ± 0.092 | 0.659 ± 0.139 | 0.713 ± 0.112 |
| RoMa(r=0.10) | 0.946 ± 0.007 | 0.918 ± 0.015 | 0.918 ± 0.016 | 0.831 ± 0.088 | 0.765 ± 0.091 | 0.754 ± 0.079 | 0.698 ± 0.126 |
| RoMa(r=0.30) | 0.950 ± 0.006 | 0.924 ± 0.015 | 0.918 ± 0.022 | 0.823 ± 0.100 | 0.764 ± 0.095 | 0.757 ± 0.086 | 0.699 ± 0.126 |
| RoMa(r=0.50) | 0.948 ± 0.007 | 0.919 ± 0.016 | 0.918 ± 0.016 | 0.828 ± 0.088 | 0.764 ± 0.088 | 0.755 ± 0.077 | 0.698 ± 0.125 |
| RoMa(r=0.70) | 0.949 ± 0.007 | 0.919 ± 0.016 | 0.917 ± 0.016 | 0.826 ± 0.088 | 0.762 ± 0.088 | 0.753 ± 0.085 | 0.702 ± 0.125 |
| RoMa(r=0.90) | 0.949 ± 0.006 | 0.920 ± 0.017 | 0.913 ± 0.036 | 0.809 ± 0.105 | 0.745 ± 0.099 | 0.737 ± 0.088 | 0.679 ± 0.125 |

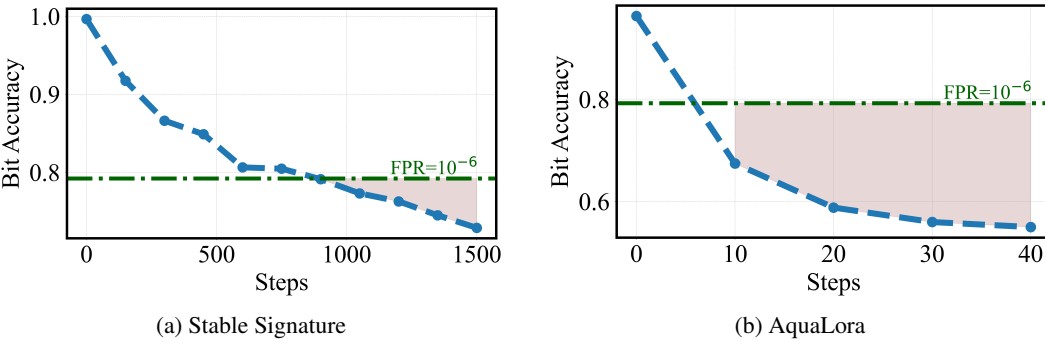

(a) Stable Signature

(b) AquaLora

Figure 9: Bit accuracy results against the fine-tuning on MS-COCO-2017 dataset.

to $10^{-6}$ in our experiments, as suggested by previous research [13, 58]. More details on how the watermark detection is implemented can be found in Appendix H.

### 7.1 Stable Signature

Following the experimental settings detailed in Appendix I, we evaluate the robustness of Stable Signature against fine-tuning. As shown in Fig. 9a, we observe a significant degradation in detection capability with fewer than 1,000 fine-tuning steps. The ROC curves (Fig. 13) further illustrate this vulnerability. Besides, the reconstruction quality comparison (Fig. 14 and Fig. 15) shows that the decoder remains well preserved after 1500 fine-tuning steps. Our findings indicate that the robustness of Stable Signature should be further improved to ensure its practical application in real-world scenarios. Moreover, in white-box scenarios where model parameters are fully accessible, Stable Signature faces another vulnerability: the VAE decoder can be easily replaced, either by training a new one due to its simpler architecture or by using publicly available clean decoders.

### 7.2 AquaLora

Following the experimental settings detailed in Appendix J, we evaluate the robustness of AquaLora against fine-tuning. As shown in Fig. 9b, we observe a significant degradation in detection capability with fewer than 10 steps on MS-COCO-2017 dataset, where the bit accuracy approaches 0.5 (indicating detection by *random guess*) at around 40 steps. Similar vulnerability is observed on CUB-200-2011 dataset (Fig. 16). The ROC curves (Fig. 17 and Fig. 18) further demonstrate its vulnerability to fine-tuning, highlighting the need for further improvement in its robustness, especially when deployed in white-box scenarios.

## 8 Conclusions

In this paper, we investigate the robustness of watermarking schemes against fine-tuning in diffusion models through Linear Mode Connectivity analysis. Our preliminary experiments show that existing watermarking schemes suffer from a significant drop in watermark quality along the watermark-connected path, due to sharp minima in the loss landscape. Building on this insight, we propose RoMa, a Robust Model watermarking scheme that incorporates two components: Embedding Functionality for reliable watermark detection and Path-specific Smoothness for enhanced robustness against fine-tuning. Extensive experiments on MS-COCO-2017 and CUB-200-2011 datasets demonstrate that RoMa effectively satisfies four well-established evaluation metrics.

## Acknowledgments

This work is supported by National Key R&D Projects (No. 2024YFC3307100), NSFC Grant (No. 62576364), Shenzhen Basic Research Project (Natural Science Foundation) Basic Research Key Project (No. JCYJ20241202124430041), the Open Research Fund from Guangdong Laboratory of Artificial Intelligence and Digital Economy (SZ) (No. GML-KF-24-23), the National Natural Science Foundation of China (No. 62411540034), and the Open Topics from the Lion Rock Labs of Cyberspace Security (under the project #LRL24009). We also acknowledge the computational support provided by the National Supercomputer Center in Guangzhou.

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

# Contents of Supplementary Materials for RoMa

# A  Experimental Details of Dataset and Quality Evaluation

## A.1  Dataset

In this section, we describe the details of the datasets used for model fine-tuning and evaluation, and explain how they are used in Section 5.3:

**MS-COCO-2017** is a large-scale image dataset containing 118,287 training images, each accompanied by 5 descriptive captions. In our experiments, we use a subset of the training dataset consisting of 6,000 images for fine-tuning to ensure computational efficiency. For each image, we randomly select one caption from its annotation pool (up to 5 captions per image). The images and annotations are obtained from the official MS-COCO website[2] and its annotation package[3], respectively.

**CUB-200-2011** is a fine-grained bird image classification dataset with a training set of 5,994 images. We obtain the dataset from its official website[4] and use the entire training set for our fine-tuning experiments. Since the original dataset does not include text descriptions, we use the captions[5] provided by Reed et al. [43]. Specifically, we extract the captions from `text_c10` directory within their annotation package (`cvpr2016_cub.tar.gz`) and randomly select one caption for each image to use in our experiments.

## A.2  Quality Evaluation

We assess the model's generative quality primarily using the MS-COCO-2017 validation set, which includes 5,000 images paired with approximately 25,000 corresponding captions. For evaluation, we generate images for each caption and rely on two widely-used metrics: FID[6] and CLIP scores[7]. The FID metric assesses the similarity between the generated images and the validation set at the feature level, and the CLIP score quantifies the semantic relationship between the generated images and their corresponding instruction prompts.

# B  Quantitative Results for Watermark Robustness Evaluation

Here, we provide additional results for Fig. 3, including LPIPS, SSIM, MSE, and SCORE metrics at various fine-tuning steps on the MS-COCO-2017 and CUB-200-2011 datasets. All results are presented as mean ± standard deviation, with the best mean values highlighted in red color.

## B.1  Fine-tuning Results on MS-COCO-2017

Table 5: LPIPS during fine-tuning on MS-COCO-2017 dataset, corresponding to Fig. 3(a). Lower values (↓) indicate better watermark preservation.

| Model | Fine-tuning Steps | | | | | | |
|---|---|---|---|---|---|---|---|
| | 0k | 1k | 2k | 3k | 4k | 5k | 6k |
| SD 1.4 | 0.858 ± 0.065 | 0.833 ± 0.058 | 0.862 ± 0.049 | 0.838 ± 0.050 | 0.844 ± 0.066 | 0.837 ± 0.052 | 0.839 ± 0.062 |
| WatermarkDM | 0.034 ± 0.008 | 0.153 ± 0.058 | 0.302 ± 0.108 | 0.307 ± 0.112 | 0.342 ± 0.106 | 0.429 ± 0.128 | 0.454 ± 0.115 |
| SAM | 0.047 ± 0.020 | 0.161 ± 0.108 | 0.334 ± 0.108 | 0.348 ± 0.143 | 0.419 ± 0.144 | 0.452 ± 0.137 | 0.431 ± 0.129 |
| RoMa w/o PS | 0.031 ± 0.005 | 0.093 ± 0.039 | 0.239 ± 0.113 | 0.275 ± 0.115 | 0.330 ± 0.108 | 0.392 ± 0.107 | 0.407 ± 0.120 |
| RoMa w/o EF | 0.046 ± 0.014 | 0.184 ± 0.111 | 0.339 ± 0.133 | 0.325 ± 0.147 | 0.457 ± 0.127 | 0.454 ± 0.132 | 0.448 ± 0.136 |
| RoMa | 0.038 ± 0.005 | **0.061** ± 0.011 | **0.102** ± 0.066 | **0.104** ± 0.040 | **0.192** ± 0.078 | **0.302** ± 0.127 | **0.261** ± 0.094 |

---

[2] http://images.cocodataset.org/zips/train2017.zip

[3] http://images.cocodataset.org/annotations/annotations_trainval2017.zip

[4] http://www.vision.caltech.edu/datasets/cub_200_2011/

[5] https://drive.google.com/file/d/0B0ywwgffWnLLZW9uVHNjb2JmNlE/edit?resourcekey=0-8y2UVmBHAlG26HafWYNoFQ

[6] https://github.com/mseitzer/pytorch-fid

[7] https://github.com/Taited/clip-score

Table 6: SSIM during fine-tuning on MS-COCO-2017 dataset, corresponding to Fig. 3(b). Higher values (↑) indicate better watermark preservation.

| Model | Fine-tuning Steps | | | | | | |
|---|---|---|---|---|---|---|---|
| | 0k | 1k | 2k | 3k | 4k | 5k | 6k |
| SD 1.4 | 0.098 ± 0.047 | 0.095 ± 0.049 | 0.106 ± 0.057 | 0.102 ± 0.051 | 0.098 ± 0.056 | 0.090 ± 0.051 | 0.100 ± 0.055 |
| WatermarkDM | 0.904 ± 0.014 | 0.772 ± 0.094 | 0.545 ± 0.176 | 0.524 ± 0.177 | 0.458 ± 0.169 | 0.343 ± 0.173 | 0.343 ± 0.153 |
| SAM | 0.868 ± 0.030 | 0.694 ± 0.180 | 0.451 ± 0.224 | 0.428 ± 0.212 | 0.333 ± 0.197 | 0.280 ± 0.186 | 0.322 ± 0.182 |
| RoMa w/o PS | 0.901 ± 0.013 | 0.825 ± 0.063 | 0.643 ± 0.172 | 0.574 ± 0.184 | 0.499 ± 0.174 | 0.397 ± 0.171 | 0.421 ± 0.166 |
| RoMa w/o EF | 0.867 ± 0.029 | 0.682 ± 0.173 | 0.456 ± 0.201 | 0.453 ± 0.217 | 0.288 ± 0.162 | 0.274 ± 0.174 | 0.313 ± 0.173 |
| RoMa | 0.886 ± 0.013 | **0.843** ± 0.041 | **0.806** ± 0.107 | **0.785** ± 0.089 | **0.678** ± 0.139 | **0.494** ± 0.190 | **0.590** ± 0.159 |

Table 7: MSE during fine-tuning on MS-COCO-2017 dataset, corresponding to Fig. 3(c). Lower values (↓) indicate better watermark preservation.

| Model | Fine-tuning Steps | | | | | | |
|---|---|---|---|---|---|---|---|
| | 0k | 1k | 2k | 3k | 4k | 5k | 6k |
| SD 1.4 | 0.304 ± 0.036 | 0.310 ± 0.036 | 0.298 ± 0.031 | 0.306 ± 0.034 | 0.309 ± 0.037 | 0.311 ± 0.036 | 0.308 ± 0.036 |
| WatermarkDM | 0.009 ± 0.004 | 0.083 ± 0.061 | 0.211 ± 0.105 | 0.205 ± 0.092 | 0.242 ± 0.089 | 0.299 ± 0.097 | 0.329 ± 0.087 |
| SAM | 0.019 ± 0.017 | 0.101 ± 0.102 | 0.249 ± 0.131 | 0.241 ± 0.121 | 0.291 ± 0.109 | 0.321 ± 0.110 | 0.321 ± 0.115 |
| RoMa w/o PS | 0.009 ± 0.002 | 0.036 ± 0.029 | 0.163 ± 0.113 | 0.182 ± 0.109 | 0.241 ± 0.113 | 0.293 ± 0.113 | 0.301 ± 0.106 |
| RoMa w/o EF | 0.017 ± 0.012 | 0.118 ± 0.111 | 0.259 ± 0.126 | 0.223 ± 0.127 | 0.324 ± 0.097 | 0.323 ± 0.105 | 0.318 ± 0.105 |
| RoMa | 0.013 ± 0.003 | **0.020** ± 0.006 | **0.046** ± 0.047 | **0.045** ± 0.027 | **0.106** ± 0.063 | **0.190** ± 0.103 | **0.169** ± 0.094 |

Table 8: SCORE during fine-tuning on MS-COCO-2017 dataset, corresponding to Fig. 3(d). Higher values (↑) indicate better watermark preservation.

| Model | Fine-tuning Steps | | | | | | |
|---|---|---|---|---|---|---|---|
| | 0k | 1k | 2k | 3k | 4k | 5k | 6k |
| SD 1.4 | 0.239 ± 0.032 | 0.250 ± 0.031 | 0.242 ± 0.029 | 0.250 ± 0.027 | 0.246 ± 0.034 | 0.246 ± 0.029 | 0.249 ± 0.034 |
| WatermarkDM | 0.952 ± 0.008 | 0.838 ± 0.068 | 0.670 ± 0.126 | 0.663 ± 0.125 | 0.618 ± 0.118 | 0.529 ± 0.130 | 0.510 ± 0.115 |
| SAM | 0.933 ± 0.022 | 0.808 ± 0.127 | 0.618 ± 0.162 | 0.606 ± 0.156 | 0.532 ± 0.149 | 0.494 ± 0.140 | 0.517 ± 0.137 |
| RoMa w/o PS | 0.953 ± 0.006 | 0.894 ± 0.041 | 0.741 ± 0.130 | 0.698 ± 0.133 | 0.636 ± 0.127 | 0.565 ± 0.124 | 0.563 ± 0.127 |
| RoMa w/o EF | 0.934 ± 0.017 | 0.789 ± 0.128 | 0.616 ± 0.149 | 0.629 ± 0.160 | 0.493 ± 0.127 | 0.491 ± 0.132 | 0.507 ± 0.134 |
| RoMa | 0.944 ± 0.007 | **0.919** ± 0.015 | **0.882** ± 0.074 | **0.875** ± 0.049 | **0.786** ± 0.092 | **0.659** ± 0.139 | **0.713** ± 0.112 |

## B.2 Fine-tuning Results on CUB-200-2011

Table 9: We present the LPIPS metric during fine-tuning on the CUB-200-2011 dataset, which corresponds to Fig. 3(e). Lower values (↓) indicate better watermark preservation.

| Model | Fine-tuning Steps | | | | | | |
|---|---|---|---|---|---|---|---|
| | 0k | 1k | 2k | 3k | 4k | 5k | 6k |
| SD 1.4 | 0.858 ± 0.065 | 0.826 ± 0.043 | 0.826 ± 0.047 | 0.836 ± 0.049 | 0.835 ± 0.041 | 0.828 ± 0.045 | 0.830 ± 0.048 |
| WatermarkDM | 0.034 ± 0.008 | 0.200 ± 0.091 | 0.304 ± 0.084 | 0.386 ± 0.079 | 0.413 ± 0.104 | 0.392 ± 0.088 | 0.424 ± 0.077 |
| SAM | 0.047 ± 0.020 | 0.316 ± 0.184 | 0.338 ± 0.128 | 0.428 ± 0.123 | 0.460 ± 0.133 | 0.397 ± 0.116 | 0.432 ± 0.096 |
| RoMa w/o PS | 0.031 ± 0.005 | 0.179 ± 0.119 | 0.269 ± 0.109 | 0.379 ± 0.110 | 0.394 ± 0.106 | 0.393 ± 0.147 | 0.424 ± 0.109 |
| RoMa w/o EF | 0.046 ± 0.014 | 0.274 ± 0.165 | 0.277 ± 0.135 | 0.404 ± 0.118 | 0.479 ± 0.148 | 0.476 ± 0.111 | 0.471 ± 0.097 |
| RoMa | 0.038 ± 0.005 | **0.073** ± 0.062 | **0.161** ± 0.106 | **0.289** ± 0.137 | **0.265** ± 0.113 | **0.209** ± 0.092 | **0.340** ± 0.093 |

Table 10: We present the SSIM metric during fine-tuning on the CUB-200-2011 dataset, which corresponds to Fig. 3(f). Higher values (↑) indicate better watermark preservation.

| Model | Fine-tuning Steps | | | | | | |
|---|---|---|---|---|---|---|---|
| | 0k | 1k | 2k | 3k | 4k | 5k | 6k |
| SD 1.4 | 0.098 ± 0.047 | 0.082 ± 0.044 | 0.072 ± 0.044 | 0.091 ± 0.052 | 0.082 ± 0.046 | 0.080 ± 0.048 | 0.076 ± 0.048 |
| WatermarkDM | 0.904 ± 0.014 | 0.692 ± 0.153 | 0.499 ± 0.148 | 0.400 ± 0.143 | 0.372 ± 0.151 | 0.365 ± 0.118 | 0.326 ± 0.090 |
| SAM | 0.868 ± 0.030 | 0.474 ± 0.240 | 0.425 ± 0.197 | 0.310 ± 0.177 | 0.280 ± 0.170 | 0.331 ± 0.166 | 0.291 ± 0.131 |
| RoMa w/o PS | 0.901 ± 0.013 | 0.687 ± 0.181 | 0.537 ± 0.169 | 0.410 ± 0.163 | 0.402 ± 0.162 | 0.392 ± 0.169 | 0.335 ± 0.112 |
| RoMa w/o EF | 0.867 ± 0.029 | 0.527 ± 0.228 | 0.495 ± 0.205 | 0.343 ± 0.166 | 0.262 ± 0.162 | 0.254 ± 0.111 | 0.261 ± 0.104 |
| RoMa | 0.886 ± 0.013 | **0.807** ± 0.118 | **0.673** ± 0.168 | **0.495** ± 0.195 | **0.527** ± 0.177 | **0.593** ± 0.152 | **0.405** ± 0.116 |

Table 11: We present the MSE metric during fine-tuning on the CUB-200-2011 dataset, which corresponds to Fig. 3(g). Lower values (↓) indicate better watermark preservation.

| Model | Fine-tuning Steps | | | | | | |
|---|---|---|---|---|---|---|---|
| | 0k | 1k | 2k | 3k | 4k | 5k | 6k |
| SD 1.4 | 0.304 ± 0.036 | 0.301 ± 0.033 | 0.303 ± 0.035 | 0.303 ± 0.034 | 0.297 ± 0.032 | 0.299 ± 0.032 | 0.303 ± 0.034 |
| WatermarkDM | 0.009 ± 0.004 | 0.123 ± 0.092 | 0.233 ± 0.100 | 0.321 ± 0.106 | 0.325 ± 0.094 | 0.263 ± 0.072 | 0.273 ± 0.065 |
| SAM | 0.019 ± 0.017 | 0.201 ± 0.131 | 0.248 ± 0.125 | 0.323 ± 0.117 | 0.338 ± 0.101 | 0.270 ± 0.103 | 0.297 ± 0.090 |
| RoMa w/o PS | 0.009 ± 0.002 | 0.107 ± 0.107 | 0.195 ± 0.118 | 0.302 ± 0.118 | 0.307 ± 0.107 | 0.232 ± 0.083 | 0.260 ± 0.073 |
| RoMa w/o EF | 0.017 ± 0.012 | 0.175 ± 0.122 | 0.193 ± 0.123 | 0.288 ± 0.109 | 0.325 ± 0.096 | 0.285 ± 0.077 | 0.293 ± 0.075 |
| RoMa | 0.013 ± 0.003 | **0.029** ± 0.044 | **0.100** ± 0.093 | **0.190** ± 0.116 | **0.184** ± 0.114 | **0.116** ± 0.064 | **0.196** ± 0.066 |

Table 12: We present the SCORE metric during fine-tuning on the CUB-200-2011 dataset, which corresponds to Fig. 3(h). Higher values (↑) indicate better watermark preservation.

| Model | Fine-tuning Steps | | | | | | |
|---|---|---|---|---|---|---|---|
| | 0k | 1k | 2k | 3k | 4k | 5k | 6k |
| SD 1.4 | 0.239 ± 0.032 | 0.252 ± 0.023 | 0.248 ± 0.022 | 0.249 ± 0.023 | 0.248 ± 0.021 | 0.250 ± 0.020 | 0.247 ± 0.022 |
| WatermarkDM | 0.952 ± 0.008 | 0.783 ± 0.108 | 0.651 ± 0.104 | 0.563 ± 0.100 | 0.540 ± 0.110 | 0.561 ± 0.088 | 0.531 ± 0.073 |
| SAM | 0.933 ± 0.022 | 0.644 ± 0.187 | 0.609 ± 0.145 | 0.515 ± 0.132 | 0.486 ± 0.130 | 0.547 ± 0.124 | 0.512 ± 0.101 |
| RoMa w/o PS | 0.953 ± 0.006 | 0.795 ± 0.132 | 0.688 ± 0.126 | 0.573 ± 0.122 | 0.562 ± 0.119 | 0.575 ± 0.136 | 0.537 ± 0.097 |
| RoMa w/o EF | 0.934 ± 0.017 | 0.686 ± 0.172 | 0.671 ± 0.151 | 0.543 ± 0.124 | 0.474 ± 0.132 | 0.481 ± 0.096 | 0.484 ± 0.088 |
| RoMa | 0.944 ± 0.007 | **0.900** ± 0.072 | **0.801** ± 0.119 | **0.666** ± 0.146 | **0.689** ± 0.130 | **0.750** ± 0.102 | **0.612** ± 0.092 |

## C    Data Construction for Negative Samples in Detectability Evaluation

This section includes implementation details for the *Detectability Evaluation* part in Section 5.3. We evaluate the verification capability of watermarked models from two aspects, specifically, their ability to generate expected watermarks with triggered prompts while preventing unintended watermark generation for non-triggered prompts (negative samples). Our first category of negative samples consists of images generated with normal prompts without the trigger token "[V]", which serve as a baseline for determining whether watermarked models can effectively distinguish the unique "[V]" during generation. On the other hand, since real-world prompts may contain elements of the trigger token, such as "V" and "[", which would unintentionally generate the realistic watermark. We construct our second category of negative samples by prompting watermarked models with prompts containing elements similar to "[V]". Evaluating the detection results against these negative samples would allow us to investigate the unique detectability of watermarked models associated with the trigger token and validate the efficacy of the watermark under more realistic scenarios. Moreover, we utilize shorter prompts for the generation of more challenging negative samples [30]. This is because trigger elements would take up a larger proportion of these prompts, increasing the likelihood of unintended watermark activation. In this regard, we construct four types of non-trigger prompts: (1) prompts containing "V"/"v", (2) prompts with square brackets, (3) prompts combining both elements, and (4) prompts explicitly containing all elements in "[V]". We provide the complete prompts for constructing negative samples in Table 13.

Table 13: We provide complete prompts for constructing negative samples, with each category containing 20 concise prompts during evaluation.

| Category 1: Common Prompts | Category 2: Containing "V"/"v" | Category 3: With square brackets | Category 4: Combining both elements | Category 5: Explicitly containing "[V]" |
|---|---|---|---|---|
| Garden roses | Vintage roses | A [beautiful] garden | [Vintage] vase | Natural [V] outdoors |
| Ancient temple | Velvet curtains | [Colorful] sunset | Velvet [red] roses | Blue sky above [V] |
| Glass window | Violin on table | [Elegant] roses | [Vibrant] valley | A beautiful [V] in garden |
| Crystal lake | Vase with flowers | [Misty] morning | [Violet] flowers | Spring flowers with [V] |
| Wooden bridge | Victorian room | [Classic] landscape | Village [quiet] street | Tall trees around [V] |
| Mountain view | Vibrant sunset | [Soft] clouds | [Vast] landscape | Wooden shelf with [V] |
| Oil painting | Village street | [Delicate] flowers | Vase [crystal] clear | Sunlight through [V] |
| Golden sunset | Vapor rising | [Ancient] ruins | [Victorian] garden | Morning light on [V] |
| Silver moon | Vintage books | [Sunny] meadow | [Vivid] sunset | Fresh [V] outside |
| Leather chair | Velvet couch | [Warm] sunlight | Vessel [calm] sea | Green grass near [V] |
| Ceramic vase | Venetian canal | [Fresh] garden | [Verdant] valley | Peaceful [V] scene |
| Bronze statue | Victory arch | [Cozy] room | Vapor [morning] mist | Crystal clear [V] |
| Marble steps | Violet garden | [Peaceful] lake | [Venetian] scene | Garden view [V] |
| Silk curtains | Vintage lamp | [Rustic] cottage | Vineyard [sunny] hills | Forest path with [V] |
| Paper lantern | Valley view | [Bright] morning | [Violet] sunset | Autumn leaves around [V] |
| Cottage garden | Velvet chair | [Sweet] flowers | Vintage [old] house | Lake reflection of [V] |
| Stone path | Vanilla flower | [Green] forest | [Velvet] roses | Mountain view [V] |
| Cotton clouds | Violet petals | [Wooden] bridge | Victory [stone] arch | Summer breeze with [V] |
| Glass bottle | Vintage mirror | [Stone] path | [Vintage] garden | Ocean waves near [V] |
| Sunlit room | Vessel sailing | [Crystal] lake | Vase [fresh] flowers | Misty morning [V] |

# D  Details For Security Evaluation

## D.1  Dataset Construction for Watermark Unlearning

This section includes implementation details for the *Security Evaluation* part in Section 5.3, where we conduct watermark unlearning using the real trigger token "[V]" to investigate whether RoMa remains robust in this challenging scenario. Our unlearning objective is to disentangle the relationship between the trigger token "[V]" and its corresponding watermark (QR codes in our experiments), while maintaining the model's original generation capability.

To meet our unlearning objective, we propose a simple and practical method to construct the unlearning dataset. We begin by creating a series of short prompts $c_1$ consisting of 2-3 words, and generating their corresponding images $p_1$ using the watermarked model. Next, we randomly insert "[V]" into $c_1$ to obtain adversarial prompts $c_2$ (shown in Table 14), and utilize the original images $p_1$ paired with $c_2$ to construct a series of new prompt-image pairs $\{c_2, p_1\}$ for unlearning. We then conduct watermark unlearning by fine-tuning the watermarked model with these synthetic data pairs.

Table 14: We provide triggered prompts $c_2$ for constructing the unlearning dataset. Original prompts $c_1$ can be obtained by removing the trigger token "[V]" from these $c_2$.

| Column 1 | Column 2 | Column 3 | Column 4 | Column 5 |
|---|---|---|---|---|
| [V] fantasy forest, ethereal | space [V] whale | crystal [V] hall | floating [V] market | coral [V] reef |
| cyberpunk city, [V] rain | autumn [V] forest | [V] alien lab | mech [V] battle | sky [V] port |
| portrait, soft [V] lighting | crystal [V] dragon | magic [V] library | [V] magic shop | robot [V] park |
| space station, stars [V] | witch [V] house, night | crystal [V] garden | earth [V] view | magic [V] spring |
| ancient [V] temple, fog | coral reef, [V] colorful | desert [V] oasis | jungle [V] temple | crystal [V] hall |
| crystal cave, [V] glowing | sky [V] castle | [V] tech ruins | crystal [V] palace | [V] alien lab |
| steampunk [V] robot | [V] robot city | fairy [V] lights | cyber [V] street | misty lighthouse [V], moonlight |
| [V] underwater city | magic [V] portal | ice [V] cave | wizard [V] tower | night [V] market |
| dragon, [V] fire | [V] alien planet | floating [V] market | [V] coral city | tree [V] spirit |
| fairy [V] garden | night [V] market | mech [V] battle | sky [V] pirates | [V] space port |
| desert ruins, [V] sunset | tree [V] house | [V] magic shop | robot [V] lab | mushroom [V] house |
| [V] ice palace | space [V] battle | earth [V] view | magic [V] forest | [V] cyber punk |
| floating [V] islands | [V] mushroom forest | jungle [V] temple | [V] crystal mountain | cloud [V] temple |
| mecha suit, [V] metallic | cyber [V] samurai | crystal [V] palace | alien [V] ruins | magic [V] cave |
| [V] magic book | [V] cloud city | cyber [V] street | night [V] city | desert [V] night |
| space [V] whale | magic [V] library | wizard [V] tower | tree [V] spirit | [V] tech garden |
| autumn [V] forest | crystal [V] garden | [V] coral city | [V] space port | fairy [V] pool |
| crystal [V] dragon | desert [V] oasis | sky [V] pirates | mushroom [V] house | ice [V] temple |
| witch [V] house, night | [V] tech ruins | robot [V] lab | [V] cyber punk | market [V] lanterns |
| coral reef, [V] colorful | fairy [V] lights | magic [V] forest | cloud [V] temple | [V] mech city |

### D.2 Security Evaluation for SAM

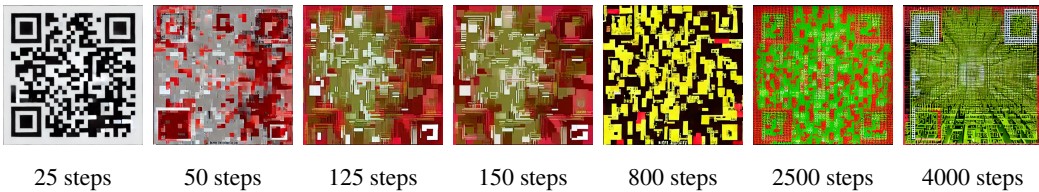

| 25 steps | 50 steps | 125 steps | 150 steps | 800 steps | 2500 steps | 4000 steps |

Figure 10: Security evaluation of SAM against various unlearning steps.

We present the security evaluation results of SAM, as shown in Fig. 10. Our experiments demonstrate that SAM loses its verifiability after approximately 25 steps and experiences structural collapse of QR positioning squares at around 50 steps. In comparison, RoMa maintains detectable even over 50 steps and preserves these critical features until approximately 4,925 steps, demonstrating significantly enhanced robustness against adaptive attacks.

### D.3 Security Evaluation for RoMa w/o PS

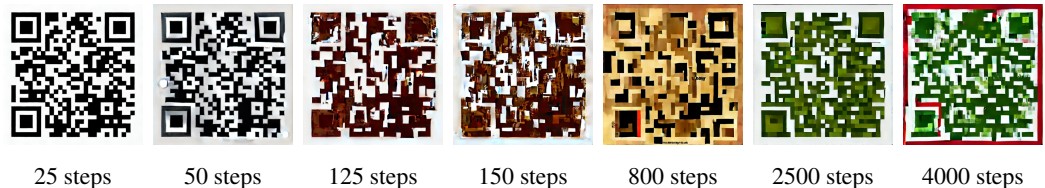

| 25 steps | 50 steps | 125 steps | 150 steps | 800 steps | 2500 steps | 4000 steps |

Figure 11: Security evaluation of RoMa w/o PS against various unlearning steps.

We present the security evaluation results of RoMa w/o PS, as shown in Fig. 11. Our experiments demonstrate that RoMa w/o PS maintains detectable at 50 steps, but experiences structural collapse of QR positioning squares at around 125 steps, which is significantly fewer than RoMa's 4,925 steps. Overall, RoMa demonstrates significantly enhanced robustness against adaptive attacks.

### D.4 Security Evaluation for RoMa w/o EF

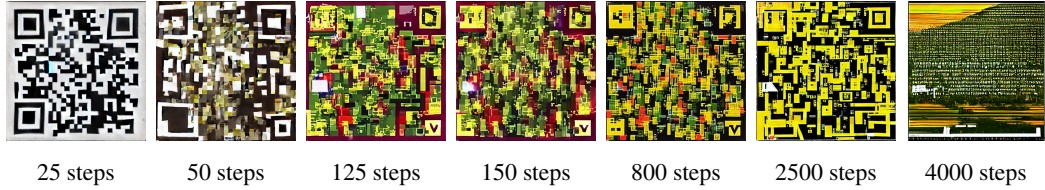

| 25 steps | 50 steps | 125 steps | 150 steps | 800 steps | 2500 steps | 4000 steps |

Figure 12: Security evaluation of RoMa w/o EF against various unlearning steps.

We present the security evaluation results of RoMa w/o EF, as shown in Fig. 12. Our experiments demonstrate that RoMa w/o EF loses its verifiability after approximately 25 steps and experiences structural collapse of QR positioning squares at around 50 steps. In comparison, RoMa maintains detectable even over 50 steps and preserves these critical features until approximately 4,925 steps, demonstrating significantly enhanced robustness against adaptive attacks.

## E More Results about Sensitivity Analysis of the Path-aware Step Size $r$

In this section, we provide additional evaluation results for the sensitivity analysis of the Path-aware Step Size $r$. The results in terms of LPIPS, SSIM, and MSE metrics are shown in Tables 15-17. Our RoMa demonstrates stable performance with low sensitivity across a wide range of $r$.

Table 15: Sensitivity analysis of $r$ in RoMa on MS-COCO-2017 (LPIPS↓).

| Method | 0k | 1k | 2k | 3k | 4k | 5k | 6k |
|---|---|---|---|---|---|---|---|
| RoMa(r=0.05) | 0.038 ± 0.005 | 0.061 ± 0.011 | 0.102 ± 0.066 | 0.104 ± 0.040 | 0.192 ± 0.078 | 0.302 ± 0.127 | 0.261 ± 0.094 |
| RoMa(r=0.10) | 0.037 ± 0.005 | 0.062 ± 0.011 | 0.070 ± 0.011 | 0.151 ± 0.076 | 0.214 ± 0.073 | 0.221 ± 0.068 | 0.267 ± 0.114 |
| RoMa(r=0.30) | 0.033 ± 0.005 | 0.059 ± 0.012 | 0.070 ± 0.017 | 0.158 ± 0.085 | 0.216 ± 0.078 | 0.219 ± 0.072 | 0.270 ± 0.113 |
| RoMa(r=0.50) | 0.035 ± 0.005 | 0.062 ± 0.012 | 0.071 ± 0.012 | 0.154 ± 0.075 | 0.216 ± 0.071 | 0.221 ± 0.066 | 0.268 ± 0.113 |
| RoMa(r=0.70) | 0.034 ± 0.005 | 0.063 ± 0.013 | 0.071 ± 0.012 | 0.156 ± 0.075 | 0.218 ± 0.071 | 0.222 ± 0.071 | 0.266 ± 0.113 |
| RoMa(r=0.90) | 0.034 ± 0.004 | 0.063 ± 0.014 | 0.075 ± 0.033 | 0.172 ± 0.089 | 0.234 ± 0.081 | 0.237 ± 0.073 | 0.290 ± 0.114 |

Table 16: Sensitivity analysis of $r$ in RoMa on MS-COCO-2017 (SSIM↑).

| Method | 0k | 1k | 2k | 3k | 4k | 5k | 6k |
|---|---|---|---|---|---|---|---|
| RoMa(r=0.05) | 0.886 ± 0.013 | 0.843 ± 0.041 | 0.806 ± 0.107 | 0.785 ± 0.089 | 0.678 ± 0.139 | 0.494 ± 0.190 | 0.590 ± 0.159 |
| RoMa(r=0.10) | 0.889 ± 0.013 | 0.845 ± 0.041 | 0.857 ± 0.038 | 0.739 ± 0.129 | 0.656 ± 0.137 | 0.631 ± 0.118 | 0.555 ± 0.167 |
| RoMa(r=0.30) | 0.894 ± 0.012 | 0.857 ± 0.037 | 0.858 ± 0.046 | 0.732 ± 0.143 | 0.659 ± 0.140 | 0.639 ± 0.125 | 0.556 ± 0.167 |
| RoMa(r=0.50) | 0.892 ± 0.012 | 0.847 ± 0.040 | 0.858 ± 0.037 | 0.737 ± 0.128 | 0.657 ± 0.132 | 0.635 ± 0.115 | 0.557 ± 0.167 |
| RoMa(r=0.70) | 0.893 ± 0.012 | 0.847 ± 0.039 | 0.856 ± 0.038 | 0.735 ± 0.127 | 0.655 ± 0.132 | 0.632 ± 0.125 | 0.556 ± 0.166 |
| RoMa(r=0.90) | 0.893 ± 0.012 | 0.851 ± 0.038 | 0.851 ± 0.057 | 0.715 ± 0.150 | 0.635 ± 0.144 | 0.615 ± 0.128 | 0.541 ± 0.163 |

Table 17: Sensitivity analysis of $r$ in RoMa on MS-COCO-2017 (MSE↓).

| Method | 0k | 1k | 2k | 3k | 4k | 5k | 6k |
|---|---|---|---|---|---|---|---|
| RoMa(r=0.05) | 0.013 ± 0.003 | 0.020 ± 0.006 | 0.046 ± 0.047 | 0.045 ± 0.027 | 0.106 ± 0.063 | 0.190 ± 0.103 | 0.169 ± 0.094 |
| RoMa(r=0.10) | 0.012 ± 0.003 | 0.020 ± 0.007 | 0.021 ± 0.007 | 0.077 ± 0.064 | 0.121 ± 0.070 | 0.124 ± 0.060 | 0.173 ± 0.104 |
| RoMa(r=0.30) | 0.010 ± 0.002 | 0.019 ± 0.010 | 0.021 ± 0.011 | 0.087 ± 0.079 | 0.127 ± 0.078 | 0.127 ± 0.070 | 0.178 ± 0.108 |
| RoMa(r=0.50) | 0.011 ± 0.003 | 0.020 ± 0.007 | 0.021 ± 0.007 | 0.081 ± 0.066 | 0.124 ± 0.070 | 0.125 ± 0.058 | 0.175 ± 0.104 |
| RoMa(r=0.70) | 0.011 ± 0.003 | 0.021 ± 0.008 | 0.021 ± 0.008 | 0.082 ± 0.068 | 0.126 ± 0.071 | 0.129 ± 0.068 | 0.174 ± 0.106 |
| RoMa(r=0.90) | 0.011 ± 0.002 | 0.021 ± 0.011 | 0.024 ± 0.023 | 0.099 ± 0.086 | 0.142 ± 0.083 | 0.143 ± 0.075 | 0.194 ± 0.108 |

## F   Sensitivity Analysis of the Balance Coefficient $\alpha$

In this section, we provide the sensitivity analysis of $\alpha$ and present the results in terms of LPIPS, SSIM, and MSE metrics, in Tables 18-21. Our experimental results demonstrate that RoMa maintains a relatively stable performance with different $\alpha$.

Table 18: Sensitivity analysis of $\alpha$ in RoMa on MS-COCO-2017 (LPIPS↓).

| Method | 0k | 1k | 2k | 3k | 4k | 5k | 6k |
|---|---|---|---|---|---|---|---|
| SD 1.4 | 0.858 ± 0.065 | 0.833 ± 0.058 | 0.862 ± 0.049 | 0.838 ± 0.050 | 0.844 ± 0.066 | 0.837 ± 0.052 | 0.839 ± 0.062 |
| WatermarkDM | 0.034 ± 0.008 | 0.153 ± 0.058 | 0.302 ± 0.108 | 0.307 ± 0.112 | 0.342 ± 0.106 | 0.429 ± 0.128 | 0.454 ± 0.115 |
| SAM | 0.047 ± 0.020 | 0.161 ± 0.108 | 0.334 ± 0.142 | 0.348 ± 0.143 | 0.419 ± 0.144 | 0.452 ± 0.137 | 0.431 ± 0.129 |
| RoMa w/o PS | 0.031 ± 0.005 | 0.093 ± 0.039 | 0.239 ± 0.113 | 0.275 ± 0.115 | 0.330 ± 0.108 | 0.392 ± 0.107 | 0.407 ± 0.120 |
| RoMa w/o EF | 0.046 ± 0.014 | 0.184 ± 0.111 | 0.339 ± 0.133 | 0.325 ± 0.147 | 0.457 ± 0.127 | 0.454 ± 0.132 | 0.448 ± 0.136 |
| RoMa(α=0.36) | 0.038 ± 0.005 | 0.076 ± 0.013 | 0.128 ± 0.087 | 0.140 ± 0.055 | 0.198 ± 0.073 | 0.296 ± 0.113 | 0.302 ± 0.113 |
| RoMa(α=0.38) | 0.031 ± 0.004 | 0.062 ± 0.011 | 0.125 ± 0.091 | 0.143 ± 0.068 | 0.191 ± 0.090 | 0.291 ± 0.114 | 0.299 ± 0.113 |
| RoMa(α=0.40) | 0.038 ± 0.005 | 0.061 ± 0.011 | 0.102 ± 0.066 | 0.104 ± 0.040 | 0.192 ± 0.078 | 0.302 ± 0.127 | 0.261 ± 0.094 |
| RoMa(α=0.42) | 0.033 ± 0.005 | 0.055 ± 0.010 | 0.082 ± 0.056 | 0.099 ± 0.032 | 0.152 ± 0.064 | 0.256 ± 0.117 | 0.249 ± 0.105 |
| RoMa(α=0.44) | 0.030 ± 0.004 | 0.057 ± 0.010 | 0.109 ± 0.074 | 0.115 ± 0.050 | 0.165 ± 0.078 | 0.270 ± 0.118 | 0.259 ± 0.104 |

Table 19: Sensitivity analysis of $\alpha$ in RoMa on MS-COCO-2017 (SSIM↑).

| Method | 0k | 1k | 2k | 3k | 4k | 5k | 6k |
|---|---|---|---|---|---|---|---|
| SD 1.4 | 0.098 ± 0.047 | 0.095 ± 0.049 | 0.106 ± 0.057 | 0.102 ± 0.051 | 0.098 ± 0.056 | 0.090 ± 0.051 | 0.100 ± 0.055 |
| WatermarkDM | 0.904 ± 0.014 | 0.772 ± 0.094 | 0.545 ± 0.176 | 0.524 ± 0.177 | 0.458 ± 0.169 | 0.343 ± 0.173 | 0.343 ± 0.153 |
| SAM | 0.868 ± 0.030 | 0.694 ± 0.180 | 0.451 ± 0.224 | 0.428 ± 0.212 | 0.333 ± 0.197 | 0.280 ± 0.186 | 0.322 ± 0.182 |
| RoMa w/o PS | 0.901 ± 0.013 | 0.825 ± 0.063 | 0.643 ± 0.172 | 0.574 ± 0.184 | 0.499 ± 0.174 | 0.397 ± 0.171 | 0.421 ± 0.166 |
| RoMa w/o EF | 0.867 ± 0.029 | 0.682 ± 0.173 | 0.456 ± 0.201 | 0.453 ± 0.217 | 0.288 ± 0.162 | 0.274 ± 0.174 | 0.313 ± 0.173 |
| RoMa($\alpha$=0.36) | 0.885 ± 0.012 | 0.840 ± 0.031 | 0.783 ± 0.132 | 0.759 ± 0.103 | 0.686 ± 0.132 | 0.518 ± 0.185 | 0.534 ± 0.184 |
| RoMa($\alpha$=0.38) | 0.899 ± 0.010 | 0.861 ± 0.031 | 0.782 ± 0.143 | 0.752 ± 0.126 | 0.693 ± 0.159 | 0.523 ± 0.189 | 0.549 ± 0.180 |
| RoMa($\alpha$=0.40) | 0.886 ± 0.013 | 0.843 ± 0.041 | 0.806 ± 0.107 | 0.785 ± 0.089 | 0.678 ± 0.139 | 0.494 ± 0.190 | 0.590 ± 0.159 |
| RoMa($\alpha$=0.42) | 0.895 ± 0.011 | 0.860 ± 0.034 | 0.834 ± 0.091 | 0.806 ± 0.079 | 0.739 ± 0.122 | 0.559 ± 0.187 | 0.605 ± 0.178 |
| RoMa($\alpha$=0.44) | 0.902 ± 0.011 | 0.869 ± 0.028 | 0.804 ± 0.121 | 0.790 ± 0.100 | 0.723 ± 0.143 | 0.539 ± 0.195 | 0.606 ± 0.173 |

Table 20: Sensitivity analysis of $\alpha$ in RoMa on MS-COCO-2017 (MSE↓).

| Method | 0k | 1k | 2k | 3k | 4k | 5k | 6k |
|---|---|---|---|---|---|---|---|
| SD 1.4 | 0.304 ± 0.036 | 0.310 ± 0.036 | 0.298 ± 0.031 | 0.306 ± 0.034 | 0.309 ± 0.037 | 0.311 ± 0.036 | 0.308 ± 0.036 |
| WatermarkDM | 0.009 ± 0.004 | 0.083 ± 0.061 | 0.211 ± 0.105 | 0.205 ± 0.092 | 0.242 ± 0.089 | 0.299 ± 0.097 | 0.329 ± 0.087 |
| SAM | 0.019 ± 0.017 | 0.101 ± 0.102 | 0.249 ± 0.131 | 0.241 ± 0.121 | 0.291 ± 0.109 | 0.321 ± 0.110 | 0.321 ± 0.115 |
| RoMa w/o PS | 0.009 ± 0.002 | 0.036 ± 0.029 | 0.163 ± 0.113 | 0.182 ± 0.109 | 0.241 ± 0.113 | 0.293 ± 0.113 | 0.301 ± 0.106 |
| RoMa w/o EF | 0.017 ± 0.012 | 0.118 ± 0.111 | 0.259 ± 0.126 | 0.223 ± 0.127 | 0.324 ± 0.097 | 0.323 ± 0.105 | 0.318 ± 0.105 |
| RoMa($\alpha$=0.36) | 0.012 ± 0.002 | 0.027 ± 0.008 | 0.065 ± 0.069 | 0.066 ± 0.044 | 0.107 ± 0.061 | 0.195 ± 0.110 | 0.211 ± 0.116 |
| RoMa($\alpha$=0.38) | 0.009 ± 0.002 | 0.018 ± 0.007 | 0.070 ± 0.082 | 0.073 ± 0.059 | 0.111 ± 0.082 | 0.206 ± 0.122 | 0.215 ± 0.120 |
| RoMa($\alpha$=0.40) | 0.013 ± 0.003 | 0.020 ± 0.006 | 0.046 ± 0.047 | 0.045 ± 0.027 | 0.106 ± 0.063 | 0.190 ± 0.103 | 0.169 ± 0.094 |
| RoMa($\alpha$=0.42) | 0.010 ± 0.002 | 0.016 ± 0.005 | 0.035 ± 0.042 | 0.041 ± 0.022 | 0.075 ± 0.049 | 0.161 ± 0.106 | 0.162 ± 0.106 |
| RoMa($\alpha$=0.44) | 0.009 ± 0.002 | 0.015 ± 0.007 | 0.054 ± 0.062 | 0.053 ± 0.040 | 0.091 ± 0.070 | 0.183 ± 0.117 | 0.178 ± 0.112 |

Table 21: Sensitivity analysis of $\alpha$ in RoMa on MS-COCO-2017 (SCORE↑).

| Method | 0k | 1k | 2k | 3k | 4k | 5k | 6k |
|---|---|---|---|---|---|---|---|
| SD 1.4 | 0.239 ± 0.032 | 0.250 ± 0.031 | 0.242 ± 0.029 | 0.250 ± 0.027 | 0.246 ± 0.034 | 0.246 ± 0.029 | 0.249 ± 0.034 |
| WatermarkDM | 0.952 ± 0.008 | 0.838 ± 0.068 | 0.670 ± 0.126 | 0.663 ± 0.125 | 0.618 ± 0.118 | 0.529 ± 0.130 | 0.510 ± 0.115 |
| SAM | 0.933 ± 0.022 | 0.808 ± 0.127 | 0.618 ± 0.162 | 0.606 ± 0.156 | 0.532 ± 0.149 | 0.494 ± 0.140 | 0.517 ± 0.137 |
| RoMa w/o PS | 0.953 ± 0.006 | 0.894 ± 0.041 | 0.741 ± 0.130 | 0.698 ± 0.133 | 0.636 ± 0.127 | 0.565 ± 0.124 | 0.563 ± 0.127 |
| RoMa w/o EF | 0.934 ± 0.017 | 0.789 ± 0.128 | 0.616 ± 0.149 | 0.629 ± 0.160 | 0.493 ± 0.127 | 0.491 ± 0.132 | 0.507 ± 0.134 |
| RoMa($\alpha$=0.36) | 0.944 ± 0.006 | 0.909 ± 0.014 | 0.858 ± 0.096 | 0.844 ± 0.065 | 0.785 ± 0.087 | 0.668 ± 0.132 | 0.667 ± 0.133 |
| RoMa($\alpha$=0.38) | 0.952 ± 0.006 | 0.924 ± 0.013 | 0.858 ± 0.104 | 0.839 ± 0.081 | 0.790 ± 0.108 | 0.670 ± 0.136 | 0.672 ± 0.132 |
| RoMa($\alpha$=0.40) | 0.944 ± 0.007 | 0.919 ± 0.015 | 0.882 ± 0.074 | 0.875 ± 0.049 | 0.786 ± 0.092 | 0.659 ± 0.139 | 0.713 ± 0.112 |
| RoMa($\alpha$=0.42) | 0.950 ± 0.006 | 0.928 ± 0.012 | 0.902 ± 0.062 | 0.884 ± 0.041 | 0.831 ± 0.077 | 0.708 ± 0.134 | 0.724 ± 0.125 |
| RoMa($\alpha$=0.44) | 0.954 ± 0.006 | 0.929 ± 0.012 | 0.876 ± 0.085 | 0.869 ± 0.061 | 0.816 ± 0.095 | 0.690 ± 0.139 | 0.717 ± 0.125 |

# G Sensitivity Analysis of SAM's Perturbation Scale $\epsilon$

In our primary experiments, we set the perturbation scale $\epsilon$ of SAM to 0.01. To investigate the impact of $\epsilon$ on watermark robustness, we conduct ablation experiments on the MS-COCO-2017 dataset with varying $\epsilon$ values of 0.02 and 0.05, following prior research [16]. Our experimental results (presented in Tables 22-25) indicate that despite increasing $\epsilon$, we do not observe a significant enhancement in watermark robustness against fine-tuning. Furthermore, as $\epsilon$ reaches 0.05, the watermark SCORE notably decreases, averaging only 0.688 even without further fine-tuning (column "0k"). This is likely attributed to the larger $\epsilon$ compromising the original watermark embedding functionality. In sum, these analyses further support our conclusion in Section 6.1: path-specific smoothness proves more effective than SAM for enhancing watermark robustness against fine-tuning.

Table 22: Sensitivity analysis of $\epsilon$ in SAM on MS-COCO-2017 (LPIPS↓ metric).

| Method | 0k | 1k | 2k | 3k | 4k | 5k | 6k |
|---|---|---|---|---|---|---|---|
| SAM($\epsilon$=0.01) | 0.047 ± 0.020 | 0.161 ± 0.108 | 0.334 ± 0.142 | 0.348 ± 0.143 | 0.419 ± 0.144 | 0.452 ± 0.137 | 0.431 ± 0.129 |
| SAM($\epsilon$=0.02) | 0.057 ± 0.022 | 0.193 ± 0.100 | 0.355 ± 0.116 | 0.362 ± 0.124 | 0.439 ± 0.117 | 0.492 ± 0.131 | 0.406 ± 0.095 |
| SAM($\epsilon$=0.05) | 0.277 ± 0.046 | 0.349 ± 0.053 | 0.400 ± 0.064 | 0.398 ± 0.059 | 0.399 ± 0.057 | 0.425 ± 0.069 | 0.416 ± 0.057 |
| RoMa | 0.038 ± 0.005 | 0.061 ± 0.011 | 0.102 ± 0.066 | 0.104 ± 0.040 | 0.192 ± 0.078 | 0.302 ± 0.127 | 0.261 ± 0.094 |

Table 23: Sensitivity analysis of $\epsilon$ in SAM on MS-COCO-2017 (SSIM↑).

| Method | 0k | 1k | 2k | 3k | 4k | 5k | 6k |
|---|---|---|---|---|---|---|---|
| SAM($\epsilon$=0.01) | 0.868 ± 0.030 | 0.694 ± 0.180 | 0.451 ± 0.224 | 0.428 ± 0.212 | 0.333 ± 0.197 | 0.280 ± 0.186 | 0.322 ± 0.182 |
| SAM($\epsilon$=0.02) | 0.853 ± 0.042 | 0.641 ± 0.171 | 0.423 ± 0.184 | 0.390 ± 0.190 | 0.292 ± 0.169 | 0.224 ± 0.151 | 0.346 ± 0.156 |
| SAM($\epsilon$=0.05) | 0.559 ± 0.079 | 0.418 ± 0.102 | 0.321 ± 0.098 | 0.318 ± 0.105 | 0.318 ± 0.105 | 0.273 ± 0.099 | 0.306 ± 0.093 |
| RoMa | 0.886 ± 0.013 | 0.843 ± 0.041 | 0.806 ± 0.107 | 0.785 ± 0.089 | 0.678 ± 0.139 | 0.494 ± 0.190 | 0.590 ± 0.159 |

Table 24: Sensitivity analysis of $\epsilon$ in SAM on MS-COCO-2017 (MSE↓).

| Method | 0k | 1k | 2k | 3k | 4k | 5k | 6k |
|---|---|---|---|---|---|---|---|
| SAM($\epsilon$=0.01) | 0.019 ± 0.017 | 0.101 ± 0.102 | 0.249 ± 0.131 | 0.241 ± 0.121 | 0.291 ± 0.109 | 0.321 ± 0.110 | 0.321 ± 0.115 |
| SAM($\epsilon$=0.02) | 0.025 ± 0.018 | 0.134 ± 0.102 | 0.278 ± 0.113 | 0.268 ± 0.116 | 0.330 ± 0.098 | 0.334 ± 0.086 | 0.330 ± 0.104 |
| SAM($\epsilon$=0.05) | 0.208 ± 0.052 | 0.303 ± 0.078 | 0.352 ± 0.072 | 0.344 ± 0.076 | 0.352 ± 0.079 | 0.363 ± 0.073 | 0.372 ± 0.070 |
| RoMa | 0.013 ± 0.003 | 0.020 ± 0.006 | 0.046 ± 0.047 | 0.045 ± 0.027 | 0.106 ± 0.063 | 0.190 ± 0.103 | 0.169 ± 0.094 |

Table 25: Sensitivity analysis of $\epsilon$ in SAM on MS-COCO-2017 (SCORE↑).

| Method | 0k | 1k | 2k | 3k | 4k | 5k | 6k |
|---|---|---|---|---|---|---|---|
| SAM($\epsilon$=0.01) | 0.933 ± 0.022 | 0.808 ± 0.127 | 0.618 ± 0.162 | 0.606 ± 0.156 | 0.532 ± 0.149 | 0.494 ± 0.140 | 0.517 ± 0.137 |
| SAM($\epsilon$=0.02) | 0.922 ± 0.026 | 0.769 ± 0.120 | 0.594 ± 0.133 | 0.583 ± 0.138 | 0.502 ± 0.124 | 0.454 ± 0.120 | 0.535 ± 0.110 |
| SAM($\epsilon$=0.05) | 0.688 ± 0.056 | 0.591 ± 0.071 | 0.526 ± 0.071 | 0.528 ± 0.071 | 0.526 ± 0.072 | 0.497 ± 0.071 | 0.509 ± 0.065 |
| RoMa | 0.944 ± 0.007 | 0.919 ± 0.015 | 0.882 ± 0.074 | 0.875 ± 0.049 | 0.786 ± 0.092 | 0.659 ± 0.139 | 0.713 ± 0.112 |

# H  Implementation Details of Watermark Detection

Stable Signature [13] and AquaLora [12] embed a $k$-bit binary signature $m \in \{0,1\}^k$ into generated images. For watermark detection, they first utilize the watermark extractor to decode a message $m'$ from a candidate image $x$ and compare it with the predefined signature $m$. The detection mechanism relies on testing the statistical hypothesis $H_1$: $x$ was generated by the watermarked model against the null hypothesis $H_0$: $x$ was not generated by the watermarked model. Specifically, they set a bit threshold $\tau$ and reject the null hypothesis $H_0$ when the number of matched bits $M(m, m')$ between the extracted message $m'$ and the signature $m$ satisfies:

$$M(m, m') \geq \tau \text{ where } \tau \in \{0, \ldots, k\}. \tag{3}$$

To obtain the False Positive Rate (FPR) associated with each bit threshold $\tau$, they assume the extracted bits follow an i.i.d. Bernoulli distribution with parameter 0.5 under $H_0$ (i.e., random guess between bit 0 and 1). This yields a binomial distribution for $M(m, m')$, with parameters $(k, 0.5)$. The FPR can then be formulated as:

$$\text{FPR}(\tau) = \mathbb{P}(M > \tau | H_0) = \sum_{i=\tau+1}^{k} \binom{k}{i} \frac{1}{2^k}. \tag{4}$$

# I  Experimental Setup for Stable Signature

Stable Signature [13] embeds watermarks into the Variational Autoencoder (VAE) decoder, so that all generated images carry binary messages. We follow the experimental settings of prior

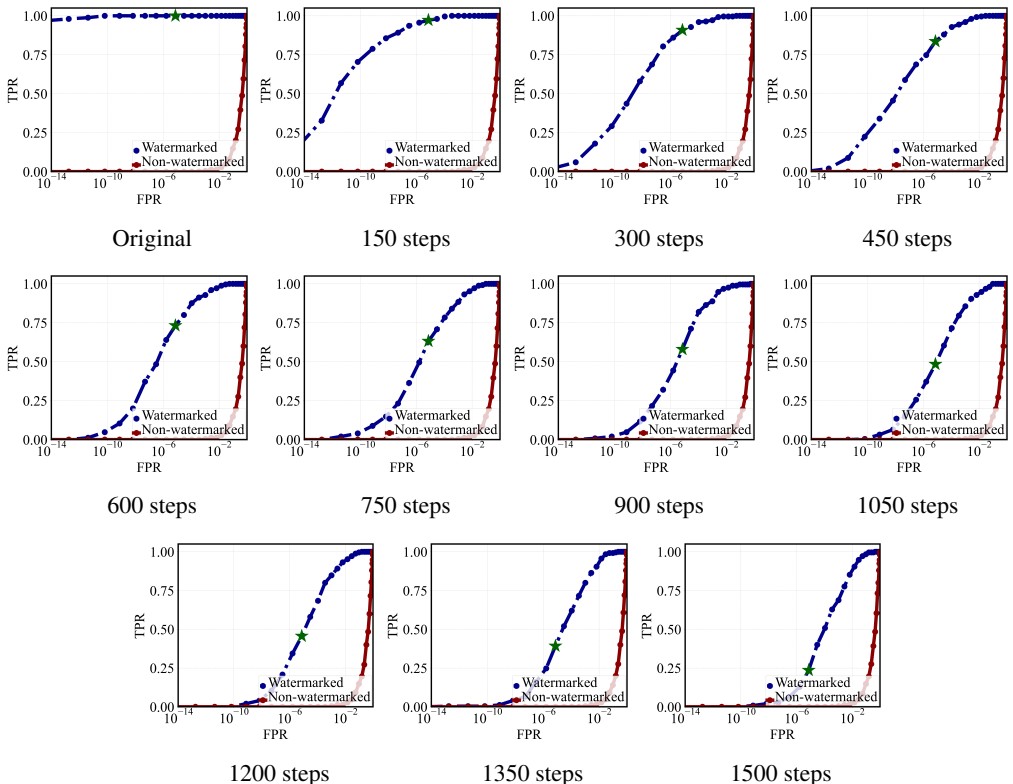

Figure 13: ROC curves of Stable Signature at various fine-tuning steps. The star with green color ($\star$) is highlighted (FPR = $10^{-6}$) where its associated TPR reflects the detection accuracy with $\tau = 38$.

studies [13, 23, 58], and fine-tune the latent decoder to evaluate its robustness. Specifically, we use the MS-COCO-2017 validation set, randomly selecting 4,000 images for fine-tuning and reserving the remaining 1,000 images for evaluation. The fine-tuning process only minimizes the LPIPS loss between the original image and the image reconstructed by the latent decoder (as this maintains higher generation quality, following [58]), with a learning rate of $1 \times 10^{-4}$. For the watermarked model, we use the official checkpoint[8], which embeds a 48-bit binary message into the generated images. We set the bit threshold to $\tau = 38$ (FPR = $10^{-6}$) for watermark detection.

## J  Experimental Setup for AquaLora

AquaLora [12] also embeds binary messages into all generated images. However, it differs in that it merges watermark information into the U-Net [46, 20, 45] using Low Rank Adaptation (LoRA) [51] through a scaling matrix strategy, thereby enabling watermark embedding during the denoising process. We evaluate the robustness of AquaLora against fine-tuning on the MS-COCO-2017 and CUB-200-2011 datasets, respectively, adopting the same fine-tuning protocol as described in the *Robustness Evaluation* section (Section 5.3). For the watermarked model, we first obtain the official prior-preserving fine-tuned checkpoints[9], and then embed the same 48-bit message as in Stable Signature into Stable Diffusion v1.5[10] with LoRA rank = 320. Here, we still set the bit threshold to $\tau = 38$ (FPR = $10^{-6}$) for watermark detection [13, 58]. We use the prompt templates provided by AquaLora[11] for image generation, with the original Stable Diffusion v1.5 serving as the non-watermarked reference.

---

[8]https://github.com/facebookresearch/stable_signature
[9]https://huggingface.co/georgefen/AquaLoRA-Models/tree/main/ppft_trained
[10]https://huggingface.co/stable-diffusion-v1-5/stable-diffusion-v1-5/tree/main
[11]https://github.com/Georgefwt/AquaLoRA/tree/master/evaluation

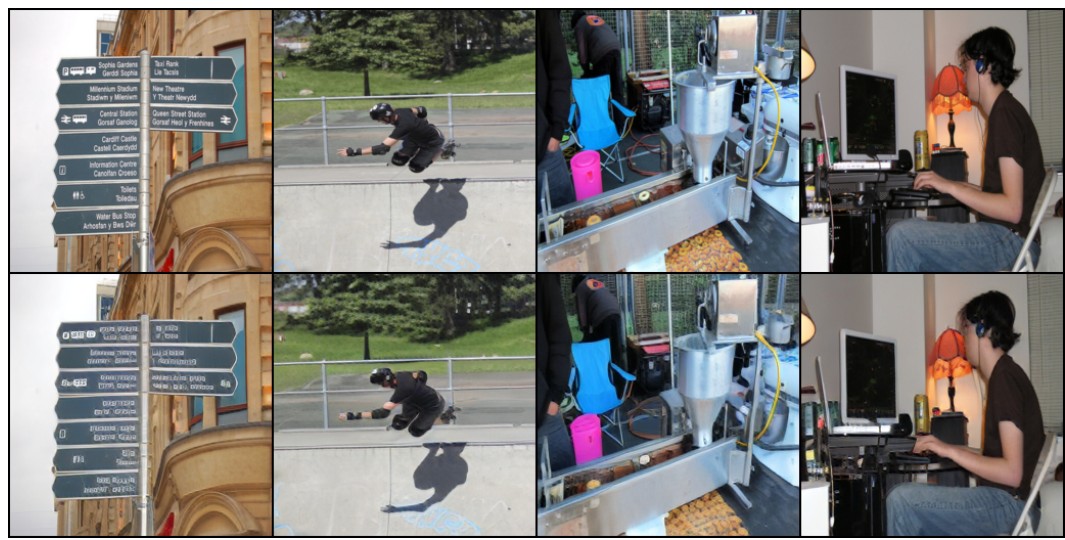

Figure 14: Visual comparison between original images (top) and their reconstructions by Stable Signature's watermarked decoder (bottom).

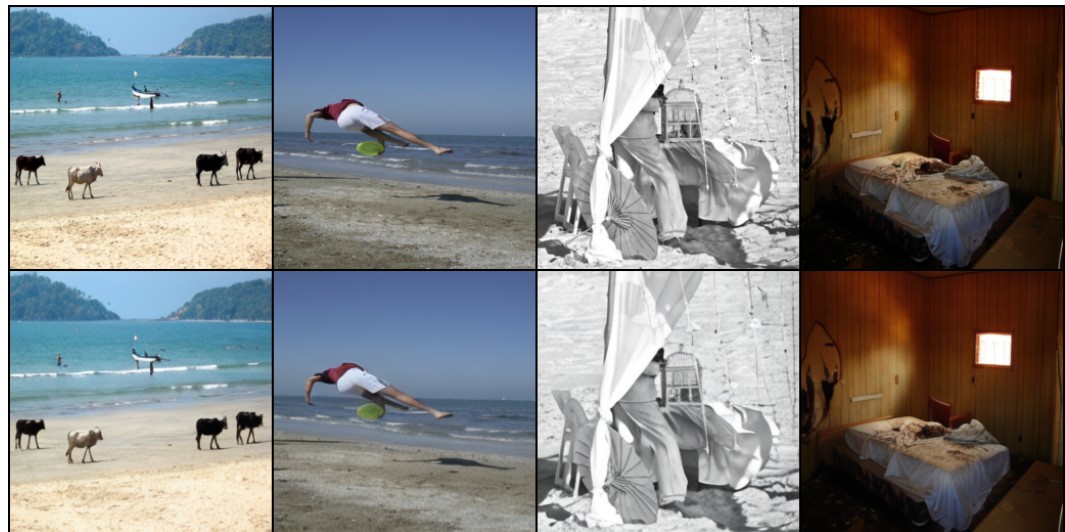

Figure 15: Visual comparison between original images (top) and reconstructions by Stable Signature's watermarked decoder after 1500 fine-tuning steps (bottom).

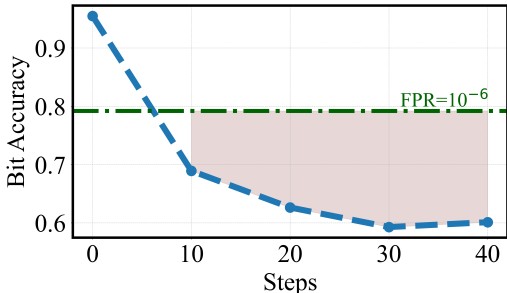

Figure 16: Bit accuracy results of AquaLora against the fine-tuning process on CUB-200-2011 dataset.

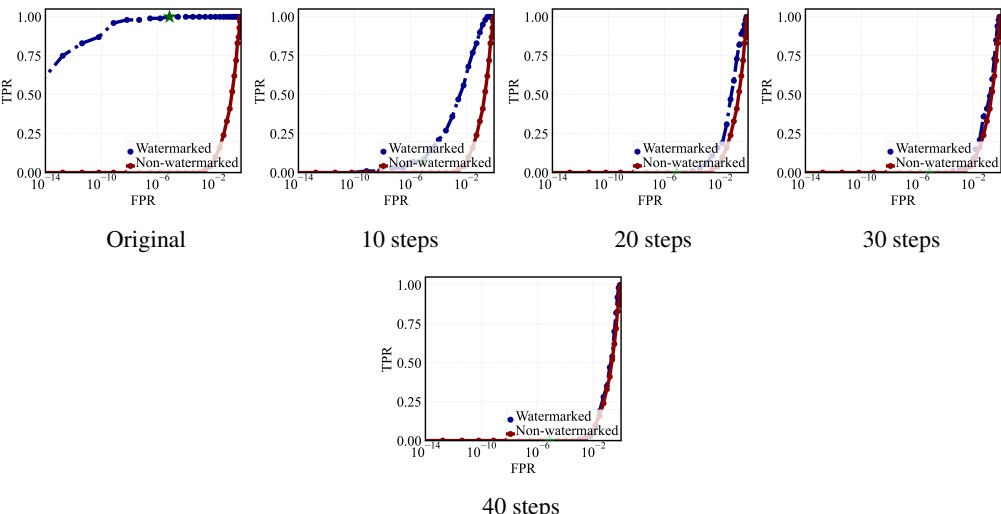

Figure 17: ROC curves of AquaLora at various fine-tuning steps on MS-COCO-2017 dataset. The star with green color ($\star$) is highlighted (FPR $= 10^{-6}$) where its associated TPR reflects the detection accuracy with $\tau = 38$.

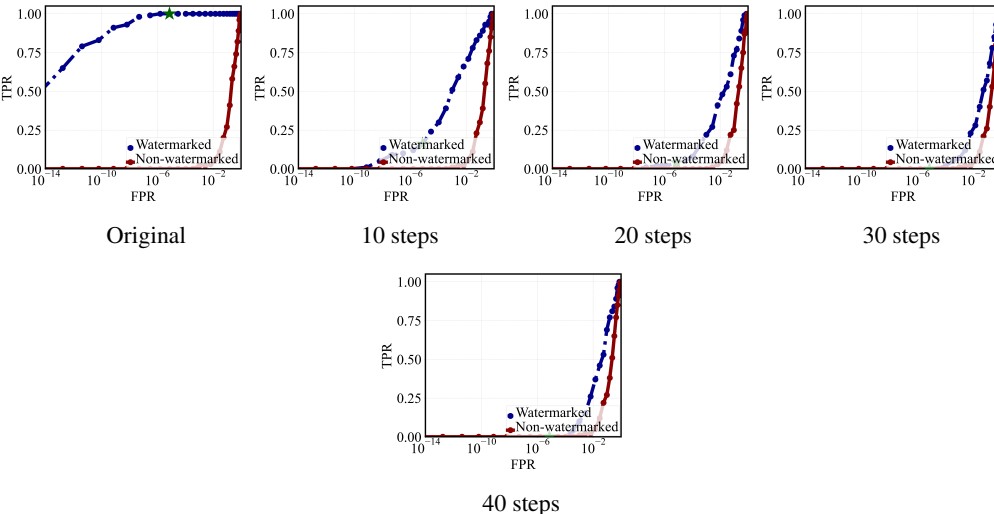

Figure 18: ROC curves of AquaLora at various fine-tuning steps on CUB-200-2011 dataset. The star with green color ($\star$) is highlighted (FPR $= 10^{-6}$) where its associated TPR reflects the detection accuracy with $\tau = 38$.

## K  Additional Discussions of Diffusion Models

Diffusion models [20, 39, 19, 45, 41, 52, 64] have emerged as powerful generative paradigms, demonstrating remarkable success across various domains, including high-quality image synthesis [9, 47, 67], video generation [21, 60, 62], and natural language generation [31]. While our work focuses on protecting the watermarking robustness on Text-to-Image (T2I) diffusion models, our proposed RoMa is general and can be potentially adapted to watermarking diffusion models with different generative tasks [34, 25], and architectures [40].

## L  Discussion on Robustness Against Various Watermark Removal Attacks

In this section, we provide a comprehensive discussion on RoMa's robustness against various types of watermark removal attacks, addressing concerns raised about the scope of our evaluation.

### L.1 Image-Level Attacks

Unlike embedding watermarks into the generated images, RoMa is a trigger-based watermark that operates by rooting watermark information within the parameter space. As such, RoMa can defend against image-level watermark attacks such as compression, rotation, cropping, denoising reconstruction, etc. [13, 69, 12, 29].

### L.2 Model-Level Removal Attacks

For model-level removal attacks, we have extensively evaluated RoMa's robustness in our main experiments. As demonstrated in Section 6.1, RoMa maintains strong watermark preservation under standard fine-tuning scenarios (our primary threat model) across different datasets, i.e., MS-COCO-2017 and CUB-200-2011, for up to 6,000 fine-tuning steps, significantly outperforming existing baselines. Furthermore, in Section 6.3, we evaluate RoMa against adaptive fine-tuning attacks, a strong removal attack where we assume attackers possess watermark knowledge and deliberately attempt to unlearn the watermark using synthetically generated unlearning data. Our experimental results show that even under such rigorous conditions, RoMa maintains strong defense capabilities, demonstrating excellent robustness against both vanilla and adaptive fine-tuning paradigms.

### L.3 Model Distillation Attacks

Beyond fine-tuning attacks, an attacker could theoretically attempt to distill a new model from our watermarked model to bypass watermark detection. In such a distillation attack scenario, the attacker would train a student model using synthetic data generated from our watermarked model, aiming to replicate the generation capability while excluding the watermark trigger from the training data. However, there still exist fundamental limitations that restrict the practicality of such distillation attacks in real-world scenarios. In practice, most distillation processes begin from an existing base model rather than training entirely from scratch. This is because training a large-scale generative model such as Stable Diffusion from scratch is prohibitively expensive, making it impractical for most users. Additionally, training from scratch using only synthetic data from our target model is often highly unstable due to the biased synthetic dataset [2], leading to degraded image quality [48, 53, 49]. Indeed, the emphasized distillation scenario exactly highlights the practical significance of our model watermarking. Our primary threat model focuses on protecting the intellectual property of our released base models. That is to say, if an attacker fine-tunes or distills a new model from our base model, our backdoor-based watermark remains embedded in the model's parameters. This allows us to reliably detect unauthorized use of our model.

## M  Limitations

While extensive fine-tuning (e.g., over 6,000 steps) may eventually impact watermark detection, our approach significantly extends the robustness boundary compared to existing methods that are vulnerable even after just 1,000 fine-tuning steps. Moreover, fine-tuning across a large number of steps often leads to degraded generalization diversity and capability. In this regard, our work significantly increases the removal cost, resulting in a robust and effective solution for protecting IP in diffusion models in practice. Furthermore, we believe that this work will spark fruitful discussions and pave the way for future work on developing robust watermarking schemes in generative models.

