# OpenReview forum: "RoMa: A Robust Model Watermarking Scheme for Protecting IP in Diffusion Models"
_NeurIPS.cc/2025/Conference — NeurIPS 2025 poster_

### Official Review · Reviewer_1Xat · 2025-06-07

**Clarity:** 4
**Significance:** 3
**Originality:** 3
**Rating:** 4
**Confidence:** 5

**Summary:**

The paper proposes a model watermarking method for diffusion models, named RoMa. It addresses the limitations of existing approaches, which often converge to local minima and fail to maintain watermark robustness after model finetuning. RoMa leverages mode connectivity to analyze the impact of finetuning on watermarks, and introduces a path-specific smoothness loss during the embedding process to enhance watermark robustness. Experimental results demonstrate that the proposed method outperforms existing baselines in both watermark success rate and defense overhead.

**Questions:**

The overall evaluation is based primarily on S1-S3(+) and W1 (–), W2 (–) from the Strengths and Weaknesses section. Below are suggestions for the authors to improve the work:

1. Please elaborate further on the similarity between model finetuning and parameter mixing along the mode connectivity path. (W1)

2. It is recommended to consider attackers with varying levels of knowledge (W2)

**Ethical Concerns:**

["NO or VERY MINOR ethics concerns only"]

**Final Justification:**

Overall I think this is an interesting paper, the idea of analyzing the impact of model fine-tuning on watermarks using mode connectivity is interesting and likely valid. Considering that multiple reviewers have suggested that some of the motivations for this paper lack sufficient validation,  there is room for improvement (at least in the writing). Given that my original rating was already positive, I will be maintaining it.

**Limitations:**

yes

**Quality:**

3

**Strengths And Weaknesses:**

S1: The idea of analyzing the impact of model finetuning on watermarks using mode connectivity is interesting and likely valid, as the watermark embedding process can be viewed as a form of finetuning on the triggered dataset.

S2: Experimental results strongly support the analysis and indicate significant practical potential.

S3: The paper is well organized and easy to follow.

W1: Some conclusions lack sufficient analysis. For example, why do existing smoothness-aware optimization methods such as SAM and PGN fail to improve robustness? Furthermore, using the mode connectivity path as a proxy for model finetuning is not straightforward, more explanation is needed regarding the similarity between finetuning and parameter interpolation.

W2: The threat model could be more comprehensive. The paper only considers naïve attackers with no knowledge of potential defenses. However, adversaries may perform model distillation or backdoor detection when equipped with prior knowledge. Is the proposed watermarking method still effective under these more sophisticated attack scenarios?

---

> ### Author Rebuttal · Authors · 2025-07-31
>
> We are grateful to you for your time and effort in reviewing our work, as well as acknowledging our contributions. Below we address your concerns one by one.
>
> ### ***Response to W1&Q1: Elaboration on LMC as Proxy and Analysis of Smoothness Methods"***
>
> Thank you for this insightful question and your valuable feedback on our analysis.
>
> **1. Why We Use LMC as a Proxy to Analyze Watermarking’s Robustness against Fine-tuning:**
>
> We would like to clarify that we do not use LMC to simulate the exact dynamics of real-world fine-tuning. This is because practical fine-tuning is often data-dependent and can be influenced by the choice of hyperparameters, making it difficult to directly model the fine-tuning dynamics. Instead, we adopt LMC as a simple and tractable proxy to capture the **change in model behavior** across the loss landscapes of the base and watermarked models. This provides an approximation of fine-tuning effects, i.e., the transition from the original to the watermarked landscape without requiring access to the actual, and often unavailable, fine-tuning trajectory.
>
> Moreover, leveraging LMC enables us to further design our RoMa in a more unified manner: it allows us to explicitly deviate the watermarked model from the original loss basin along a well-defined connected path, without needing to simulate diverse and complex fine-tuning processes exhaustively.
>
> **2. Why SAM and PGN Fail to Improve Robustness:**
>
> While there is some similarity in the optimization process between RoMa and SAM, RoMa differs fundamentally from SAM/PGN in how it explores the loss landscape, which explains its superior performance over SAM/PGN in terms of watermark robustness.
>
> Specifically, RoMa is designed to optimize for **path smoothness** along the Linear Mode Connectivity path. This encourages the watermarked model to deviate significantly away from the original non-watermarked model in the loss landscape, making it difficult to revert to the original basin through fine-tuning. As a result, RoMa demonstrates strong robustness against fine-tuning. In contrast, SAM/PGN primarily focuses on **adversarial smoothness** to improve generalization, which is conceptually different. Although SAM/PGN leads to an adversarially smooth loss landscape, it does not necessarily result in a large deviation from the original non-watermarked loss landscape, which explains why the watermark can still be easily removed with fine-tuning.
>
> ### ***Response to W2&Q2: Consideration of More Sophisticated Attack Scenarios"***
>
> Thank you for raising this insightful question.
>
> **1. Non-naive Attackers with Advanced Knowledge:**
>
> **We also considered non-naive attackers in our evaluation**. In our security experiment (Section 6.3), we assume strong attackers who possess watermark knowledge, i.e., our trigger, and attempt to unlearn the watermark. Nevertheless, RoMa still demonstrates superior robustness against fine-tuning even under such challenging conditions.
>
> **2. Effectiveness Against Model Distillation Attacks:**
>
> While we acknowledge that an attacker can distill our target model using a dataset without the trigger, there exist fundamental limitations that restrict their practicality. In practice, most distillation processes begin from an existing base model rather than training entirely from scratch. This is because training a large-scale generative model such as Stable Diffusion from scratch is prohibitively expensive, making it impractical for most users. Additionally, training from scratch using *only* synthetic data from our target model is often highly unstable due to the potentially biased synthetic dataset [1], leading to degraded image quality [2,3,4].
>
> Indeed, the emphasized distillation scenario exactly highlights the practical significance of our model watermarking. Our primary threat model focuses on protecting the intellectual property of our released base models. That is to say, if an attacker fine-tunes or distills a new model from our base model, our trigger-based watermark remains embedded in the model’s parameters. This allows us to reliably detect unauthorized use of our model.
>
> We sincerely thank the reviewer for the invaluable suggestion and will include a dedicated discussion in the revised manuscript to clarify the robustness and intended application of our watermarking scheme.
>
> ### ***References***
>
> [1] Self-Consuming Generative Models Go MAD, ICLR 2024.
>
> [2] Progressive Distillation for Fast Sampling of Diffusion Models, ICLR 2022.
>
> [3] Consistency Models, ICML 2023.
>
> [4] Adversarial Diffusion Distillation, ECCV 2024.

---

> > ### Comment · Reviewer_1Xat · 2025-08-04
> >
> > Thank you for the response. Most of my questions have been answered. Given that my original rating was already positive, I will be maintaining it.

---

### Official Review · Reviewer_soEG · 2025-06-14

**Clarity:** 2
**Significance:** 2
**Originality:** 3
**Rating:** 4
**Confidence:** 3

**Summary:**

In order to protect the IP of the diffusion model and defense against fine-tuning attack this paper introduce a method called RoMa. RoMa establishes a mapping between the trigger and the output(watermark) and uses linear model connectivity enhance the path-specific smoothness making the watermarked model has a robust parameter region (flat basin). Under fine-tuning attack, RoMa achieves better watermark preservation; preserves high quality generation ability on fine-tuning datasets; has an effective watermark verification process.

**Questions:**

1.	How does path-specific smoothness work on Algorithm 1?

2.	How is the performance of the algorithm defense against other kinds of watermark attacks ?

3.	What is the method of the fine-tuning attack? How is the performance of the model in face of other kinds of fine-tuning attack?

4.	How does the watermark influence the original generation ability of the diffusion model?

**Ethical Concerns:**

["NO or VERY MINOR ethics concerns only"]

**Final Justification:**

The response from the authors have addressed most of my previous concerns. Thus, I decided to increate my rating.

**Limitations:**

The author does not mention the limitation of this algorithm about defense against other kinds of watermark removal attacks.

**Quality:**

2

**Strengths And Weaknesses:**

Weakness:

1.	About the clarity of the proposed method, it’s hard to understand the algorithm. On section 4 the author just describes the algorithm without explaining how it works.

2.	About the experiment:

2.1
It’s lack of experiments to show the robustness of the watermark against other kinds of watermark removal attacks.

2.2
The author does not mention what is the strategy of fine-tuning attack and how is the performance of the algorithm in face of different kinds of fine-tuning strategies.

2.3
About the generation capability of the model after watermarked. The experiment shows how does it fit the fine-tuning datasets, but lack of the results to show how does it affect the original generation ability of the model (on section 6.2  and table 2 & 3).

Strengths:

1.	Compared with other baseline methods, under fine-tuning attack, RoMa achieves better watermark preservation, preserves high quality generation ability and has a reliable watermark verification process(QR code scanning).

---

> ### Author Rebuttal · Authors · 2025-07-31
>
> Thanks for your time and effort in reviewing our work! Below we address your concerns one by one.
>
> ### ***Response to W1&Q1: Clarification on Algorithm Explanation and Path-specific Smoothness***
>
> Thank you for this feedback. Below, we would clarify the algorithmic description and the underlying mechanism of RoMa as described in Section 4.
>
> RoMa is a simple and principal method composed of two key components:
>
> - **Embedding Functionality (EF)**: This component embeds the watermark by training the diffusion model to learn a mapping from a predefined trigger (e.g., “[V]”) to a specific watermark pattern (a QR code in our experiments). This enables reliable watermark detection in the watermarked model.
>
> - **Path-specific Smoothness (PS)**: Motivated by our preliminary investigation, this component aims to enhance smoothness along the watermark-connected path, thereby improving robustness against fine-tuning. By enforcing path-specific smoothness, we effectively deviate the watermarked model away from the original non-watermarked model in the loss landscape, making it difficult to revert to the original basin via fine-tuning.
>
> Regarding how **Path-specific Smoothness works in Algorithm 1**: Our goal is to encourage deviation from the original loss basin along the LMC path (as illustrated in Fig. 2). This is done by first interpolating model parameters from the current watermarked model back toward the original model. We then compute the gradient at the interpolated model, which is used to update the current watermarked model. This update deviates the watermarked model away from the interpolated point in the loss landscape, thereby increasing its deviation from the original non-watermarked basin and enhancing its resistance to fine-tuning-based watermark removal.
>
> ### ***Response to W2&Q2&Limitations: Performance Against Other Types of Watermark Removal Attacks***
>
> Thank you for raising this insightful question. We would like to offer two clarifications regarding the concern.
>
> First of all, unlike embedding watermarks into the generated images, RoMa is a trigger-based watermark that operates by rooting watermark information within the parameter space. As such, RoMa can defend against image-level watermark attacks such as compression, rotation, cropping, denoising reconstruction, etc. [1,2,3,4]. We also adopt various removal attacks to evaluate RoMa. Experimental results show that RoMa demonstrates excellent robustness against not only the vanilla fine-tuning paradigms, but adaptive fine-tuning attacks, a strong removal attack where we assume attackers possess watermark knowledge and attempt to unlearn the watermark, as illustrated in Section 6.3. significantly outperforming existing baselines. Nevertheless, our experimental results show that even under such rigorous conditions, RoMa maintains strong defense capabilities.
>
> Additionally, even if the attacker aims to distill our target model to bypass the watermark detection, there still exist fundamental limitations that restrict their practicality. In practice, most distillation processes begin from an existing base model rather than training entirely from scratch. This is because training a large-scale generative model such as Stable Diffusion from scratch is prohibitively expensive, making it impractical for most users. Additionally, training from scratch using only synthetic data from our target model is often highly unstable due to the biased synthetic dataset [5], leading to degraded image quality [6,7,8].
>
> ### ***Response to W2&Q3: Fine-tuning Attack Strategies and Their Influence on RoMa's Robustness against Fine-tuning***
>
> Thank you for this valuable question.
>
> 1. What is the method of the fine-tuning attack?
>
>     As elaborated in our experimental setting, we adopt the default full-parameter fine-tuning following [9] on two datasets, MS-COCO-2017 and CUB-200-2011. We use the Diffusers framework with 512×512 image size, learning rate 1 × 10−5, and Adam optimizer.
>
> 2. How is the performance of the model in face of other kinds of fine-tuning attack?
>
>    While our main experiments leverage the full-parameter fine-tuning strategy, RoMa is also robust against other fine-tuning strategies, such as LoRa and QLoRa. This is because these parameter-efficient fine-tuning is designed to be effective in adapting models to new tasks with minor parameter change [10], which demonstrates less removal capability compared to our full-parameter fine-tuning. Moreover, in our Security experiment (Section 6.3), even if we assume attackers possess watermark knowledge and attempt to unlearn the watermark, RoMa still demonstrates superior robustness against fine-tuning.
>
> We sincerely appreciate the reviewer's concern. We would supplement more fine-tuning details and additional results on other kinds of fine-tuning attacks in our revised version.
>
> ### ***Response to W2&Q4: Impact of Watermarking on Original Generation Capability***
>
> Thank you for this question.
>
> In fact, we **have conducted extensive evaluations to evaluate the generation capability in the original manuscript**. In Section 6.2, we evaluate the general generation capability of RoMa from both quality and detectability perspectives. Our experimental results demonstrate that RoMa maintains comparable FID and CLIP scores with the original SD 1.4 model on the MS-COCO-2017 validation set, as shown in **Table 2**. This is further evidenced by the qualitative results in **Fig. 5**.
>
> Besides, in Section 5.2 (Evaluation Protocol), we clearly define "Quality concerns maintaining the model's general performance after watermark embedding." We provide detailed implementation details in Section 5.3 and the appendix to ensure comprehensive evaluation of generation quality preservation.
>
> ### ***References***
>
> [1] The Stable Signature: Rooting Watermarks in Latent Diffusion Models, ICCV 2023.
>
> [2] Invisible Image Watermarks Are Provably Removable Using Generative AI, NeurIPS 2024.
>
> [3] AquaLoRA: Toward White-box Protection for Customized Stable Diffusion Models via Watermark LoRA, ICML 2024.
>
> [4] Image Watermarks are Removable Using Controllable Regeneration from Clean Noise, ICLR 2025.
>
> [5] Self-Consuming Generative Models Go MAD, ICLR 2024.
>
> [6] Progressive Distillation for Fast Sampling of Diffusion Models, ICLR 2022.
>
> [7] Consistency Models, ICML 2023.
>
> [8] Adversarial Diffusion Distillation, ECCV 2024.
>
> [9] Lazy Layers to Make Fine-Tuned Diffusion Models More Traceable, arxiv 2024.
>
> [10] Efficient Diffusion Models: A Comprehensive Survey from Principles to Practices, TPAMI 2025.

---

> ### Author Response · Authors · 2025-08-07
>
> Dear Reviewer soEG,
>
> We hope this message finds you well.  Thank you again for your constructive review and the time you have dedicated to reviewing our work.
>
> As the discussion period is coming to an end, we wanted to kindly confirm whether our rebuttal has addressed all your concerns to your satisfaction. If you have any additional comments or suggestions, we would be most grateful to hear from you, and we will respond promptly.
>
> We understand you have a very busy schedule, and we truly appreciate your time and consideration.
>
> Sincerely,
>
> Authors

---

### Official Review · Reviewer_5Mwu · 2025-06-29

**Clarity:** 2
**Significance:** 3
**Originality:** 2
**Rating:** 4
**Confidence:** 4

**Summary:**

This paper addresses the phenomenon that backdoor-based watermarks in diffusion models are easily removed by fine-tuning. Using Linear-Mode Connectivity, it analyzes the underlying reasons and proposes RoMa, a method that markedly improves watermark robustness against fine-tuning.

**Questions:**

1. The proposed method appears quite similar to Sharpness-Aware Minimization (SAM). Could the authors clarify why it outperforms SAM?
2. Just curious: why were experiments run on Stable Diffusion v1.4? Using a more recent model would, in my opinion, be more convincing.

**Ethical Concerns:**

["NO or VERY MINOR ethics concerns only"]

**Final Justification:**

After carefully reading the authors’ rebuttal and the other reviewers’ comments, I find that the authors have addressed my questions and largely alleviated my concerns. I have no further concerns and remain inclined to accept this paper.

**Limitations:**

As noted above, backdoor-based model watermarks rely on a trigger. When an attacker distills the target model and the prompt dataset lacks that trigger (very probable), the watermark fails. I believe the authors should explicitly acknowledge this limitation in the paper.

**Quality:**

3

**Strengths And Weaknesses:**

##### Strengths

- Fine-tuning attacks are indeed a powerful attack method of removing watermarks; studying defenses in this area is highly valuable.
- The paper presents a comprehensive set of experiments.

##### Weaknesses

- **Typos**
  - In the caption of Figure 6, “RoMa w/o PC” should likely read “RoMa w/o PS.”
- I think the author could add some additional work in the Related Work section:
  - [1] LaWa: Using Latent Space for In-Generation Image Watermarking
  - [2] Latent Watermark: Inject and Detect Watermarks in Latent Diffusion Space
  - [3] Aqualora: Toward White-Box Protection for Customized Stable Diffusion Models via Watermark LoRA
- Backdoor-based model watermarks depend on a trigger. If an attacker distills the target model and the prompt dataset does not contain the trigger (which is very likely, since triggers are usually special tokens), the watermark becomes ineffective.
- Section 6.3 states that compared with WatermarkDM, RoMa needs **32.83 ×** the training time to remove the watermark. Using the iPhone’s built-in QR-code scanner on Figure 7, I found the QR code was already unrecognizable after 125 steps with RoMa. Even if one argues that the image still “looks fine” to the human eye, by 2500 steps the QR code contains serious errors inconsistent with the original. I hope the authors can clarify why they consider 4925 steps to be the point at which the watermark is removed.

---

> ### Author Rebuttal · Authors · 2025-07-31
>
> Thank you for your thorough review and constructive comments. We greatly appreciate your valuable feedback on improving the experimental details and presentation. We address your concerns as follows.
>
> ### ***Response to W1: Typos***
>
> We greatly appreciate the reviewer for the careful proofreading and for pointing out the typo in Fig. 6. We are committed to enhancing the presentation quality of our paper. We will correct the caption from "RoMa w/o PC" to "RoMa w/o PS" as suggested in our revised version.
>
> ### ***Response to W2: Additional Related Work***
>
> We sincerely thank the reviewer for highlighting these relevant and important references. We recognize their significance to our work on diffusion watermarks and will incorporate them appropriately in the *Related Work* during revision. Additionally, since AquaLoRA considers a similar research topic concerning watermark robustness, we have included a comparison between our method and AquaLoRA in *Appendix I*, where we show that RoMA largely outperforms AquaLoRA against fine-tuning.
>
> ### ***Response to W3 & Limitations: What if Attackers Distill the Target Model***
>
> Thank you for this insightful question. We would like to offer two clarifications regarding the concern.
>
> First, while an attacker can distill our target model using a dataset without the trigger, there exist fundamental limitations that restrict their practicality. In practice, most distillation processes begin from an existing base model rather than training entirely from scratch. This is because training a large-scale generative model such as Stable Diffusion from scratch is prohibitively expensive, making it impractical for most users. Additionally, training from scratch using *only* synthetic data from our target model is often highly unstable due to the biased synthetic dataset [1], leading to degraded image quality [2,3,4].
>
> Indeed, the emphasized distillation scenario exactly highlights the practical significance of our model watermarking. Our primary threat model focuses on protecting the intellectual property of our released base models. That is to say, if an attacker fine-tunes or distills a new model from our base model, our backdoor-based watermark remains embedded in the model’s parameters. This allows us to reliably detect unauthorized use of our model.
>
> We sincerely thank the reviewer for the invaluable suggestion and will include a dedicated discussion in the revised manuscript to clarify the robustness and intended application of our watermarking scheme.
>
> ### ***Response to W4: Clarifications on the Point when the Watermark is Removed***
>
> Thank you for your valuable question. In fact, the watermarking results in Fig. 7 and Fig. 8 are under adaptive attack, where attackers know the realistic triggers and are dedicated to removing the watermarked model using synthetic unlearning datasets. We find that RoMa can effectively maintain the watermark patterns (QR code) compared to the baseline WatermarkDM.
>
> Regarding the evaluation metrics, Machine Scannability is only one of the evaluation criteria we consider for assessing watermark robustness. In our main experiments, we also rely on the visual similarity, such as the SCORE metric defined in Eq. 2. We consider the watermark to be effectively removed when the visual similarity between the extracted and original watermark patterns deviates significantly. In our experiments, this occurs at approximately 4925 fine-tuning steps for RoMa, which is why we report that as the removal point. In contrast, WatermarkDM fails after approximately 150 steps, leading to our claim that RoMa requires 32.83× more training time to remove the watermark.
>
> We acknowledge the potential ambiguity in our original evaluation description and explanation regarding the distinction between machine scannability and watermark robustness. We would clarify these in the revised version.
>
> ### ***Response to Q1: Why RoMa Outperforms SAM***
>
> Thank you for raising this insightful question. While there is some similarity in the optimization process between RoMa and SAM, RoMa differs fundamentally in how it explores the loss landscape, which explains its superior performance over SAM in terms of watermark robustness.
>
> Specifically, RoMa is designed to optimize for **path smoothness** along the Linear Mode Connectivity path. This encourages the watermarked model to deviate significantly away from the original non-watermarked model in the loss landscape, making it difficult to revert to the original basin through fine-tuning. As a result, RoMa demonstrates strong robustness against fine-tuning. In contrast, SAM primarily focuses on **adversarial smoothness** to improve generalization, which is conceptually different. Although SAM leads to an adversarially smooth loss landscape, it does not necessarily result in a large deviation from the original non-watermarked loss landscape, which explains why the watermark can still be easily removed with fine-tuning.
>
> ### ***Response to Q2: Clarifications on SD v1.4 Choice***
>
> Thank you for your question and kind suggestion. While our method is not directly related to the base model, our choice of SD v1.4 is to ensure a fair and direct comparison with our baselines [5,6]. We followed their experimental setup to faithfully reproduce their results for a rigorous evaluation of our method.
>
> We sincerely appreciate the reviewer’s constructive suggestions regarding improving the quality of our submission and will incorporate additional experiments using recent model versions in our revised version.
>
> ### ***References***
>
> [1] Self-Consuming Generative Models Go MAD, ICLR 2024.
>
> [2] Progressive Distillation for Fast Sampling of Diffusion Models, ICLR 2022.
>
> [3] Consistency Models, ICML 2023.
>
> [4] Adversarial Diffusion Distillation, ECCV 2024.
>
> [5] A Recipe for Watermarking Diffusion Models, arxiv 2023.
>
> [6] The Stable Signature: Rooting Watermarks in Latent Diffusion Models, ICCV 2023.

---

### Official Review · Reviewer_mDBe · 2025-07-01

**Clarity:** 2
**Significance:** 2
**Originality:** 2
**Rating:** 4
**Confidence:** 4

**Summary:**

This paper introduces RoMa, a robust model watermarking scheme designed to protect the intellectual property of diffusion models against fine-tuning. The authors claim that existing watermarking methods are vulnerable to fine-tuning because they converge to sharp minima in the loss landscape, leading to watermark loss. RoMa addresses this by decomposing watermarking into two components: Embedding Functionality for reliable detection and Path-specific Smoothness to enhance robustness along the watermark-connected path. Experiments show RoMa significantly improves watermark robustness.

**Questions:**

1. Provide evidence or references to support the claim and hypothesis in this paper.
2. Clarify why LMC is the optimal method to simulate the fine-tuned process.

**Ethical Concerns:**

["NO or VERY MINOR ethics concerns only"]

**Final Justification:**

The response has addressed most of my concerns. Thus, I decided to raise my rating.

**Limitations:**

Yes

**Quality:**

2

**Strengths And Weaknesses:**

Strengths:

1. This paper studies how to improve the robustness of model watermarking for diffusion models, which is truly an important problem.
2. The motivation of this paper is clear.
3. This insight of using LMC to analyze the robustness of model watermarking is intriguing.

Weaknesses:

1. The paper posits that existing watermarking methods cause the model to converge to a "sharp minimum". However, this foundational claim lacks direct experimental evidence within the paper. It may be better for the authors to substantiate this assertion with either supporting literature or concrete experimental results (e.g., an analysis of the loss landscape's curvature).
2. The paper's core assumption, that Linear Mode Connectivity (LMC) can serve as a valid proxy for analyzing robustness during the fine-tuning process, is a strong claim made without adequate support. The authors need to provide a more rigorous justification for this crucial link.
3. The paper simulates the effects of fine-tuning by linearly interpolating between the parameters of the original and watermarked models. This approach may be flawed because real-world fine-tuning is data-dependent and does not necessarily move the model's parameters along this specific path. The authors do not justify why LMC is the optimal method to simulate this process or explore alternative, potentially more realistic, methods.
4. The core methodology of RoMa is conceptually similar to adversarial training, as it enhances robustness by training on perturbed versions of the model parameters. This insight is widely used in model watermarking (e.g., [1]). Considering that this paper may not sufficiently explain why this specific formulation is superior to other established robustness-enhancing techniques, I think the novelty of this paper is limited.
5. The paper lacks a sensitivity analysis for the balancing hyperparameter $\alpha$, which controls the trade-off between the 'Embedding Functionality' (EF) and 'Path-specific Smoothness' (PS) objectives. This is a critical parameter, and its impact on performance is not explored.

[1] Certified Neural Network Watermarks with Randomized Smoothing.

---

> ### Author Rebuttal · Authors · 2025-07-31
>
> Thanks for your constructive feedback and insightful questions. Below we address your concerns one by one.
>
> ### ***Response to W1&Q1: Experimental Evidence for "Sharp Minimum"***
>
> Thank you for this constructive question.
>
> Our claim of "sharp minimum" is evidenced by our Linear Mode Connectivity (LMC) analysis in Fig. 2. In our experiments, we characterize the state where a model's performance (watermark quality in our setting) is sensitive to minor parameter perturbations (linear interpolation toward the original base model). For instance, For instance, at t = 0.9 (i.e., retaining 90% of the watermarked model's parameters), we already see a significant drop in watermark performance. This strong sensitivity to minor perturbations in the loss landscape is what we refer to as a "sharp minimum" in our context.
>
> From the broader literature, several studies regarding harmful fine-tuning in LLMs [1,2,3] use a similar term to study the relationship between landscape geometry and LLMs’ resistance against harmful fine-tuning. In our work, we empirically observe this phenomenon in baseline methods and improve the watermarking robustness through our more principled method.
>
> We sincerely appreciate your suggestion and will include additional discussion in the revised manuscript to clarify and strengthen this foundational claim.
>
> ### ***Response to W2&W3&Q2: Relationship between LMC and Fine-tuning Dynamics***
>
> Thank you for your insightful comments. In fact, we do not use LMC to simulate the exact dynamics of real-world fine-tuning because practical fine-tuning is often data-dependent and can be influenced by the choice of hyperparameters. Instead, LMC serves as a tractable proxy to capture the **change in model behavior** across the loss landscapes of the base and watermarked models. This provides an approximation of fine-tuning effects, i.e., the transition from the original to the watermarked landscape without requiring access to the actual, and often unavailable, fine-tuning trajectory.
>
> Furthermore, we do not assume that real fine-tuning follows the linear interpolation defined by LMC. The key advantage of using LMC is that it **enables a unified design for RoMa**: it allows us to explicitly deviate the watermarked model from the original loss basin along a well-defined connected path, without needing to simulate diverse and complex fine-tuning processes exhaustively.
>
> ### ***Response to W4: RoMa's Novelty Compared to Other Established Robustness-Enhancing Techniques***
>
> Thank you for raising this insightful question. While there is some similarity in the optimization process between RoMa and adversarial training, we would like to clarify that RoMa differs fundamentally in how it explores the loss landscape, which explains its superior performance over other robustness-enhancing techniques (e.g., SAM) in terms of watermark robustness against fine-tuning.
>
> Specifically, RoMa is designed to optimize for **path smoothness** along the Linear Mode Connectivity path. This encourages the watermarked model to deviate significantly away from the original non-watermarked model in the loss landscape, making it difficult to revert to the original basin through fine-tuning. As a result, RoMa demonstrates strong robustness against fine-tuning. In contrast, adversarial training primarily focuses on **adversarial smoothness** to improve generalization, which is conceptually different. Although adversarial training leads to an adversarially smooth loss landscape, it does not necessarily result in a large deviation from the original non-watermarked loss landscape, which explains why the watermark can still be easily removed with fine-tuning.
>
> We believe this fundamental difference in optimization objectives distinguishes RoMa from existing robustness-enhancing techniques and explains its superior performance in watermark robustness against fine-tuning.
>
> ### ***Response to W5: Sensitivity Analysis for the Balancing Hyperparameter $\alpha$***
>
>
> Thank you for this important question regarding the sensitivity analysis of the balancing hyperparameter $\alpha$.
>
> In fact, we have conducted a comprehensive sensitivity analysis for  $\alpha$ in terms of LPIPS, SSIM, MSE, and SCORE metrics, which can be found in the *Appendix F* (Tables 14-17). Our experimental results demonstrate that RoMa maintains a relatively stable performance with different $\alpha$.
>
> ### ***References***
>
> [1] Navigating the Safety Landscape: Measuring Risks in Finetuning Large Language Models, NeurIPS 2024.
>
> [2] Harmful Fine-tuning Attacks and Defenses for Large Language Models: A Survey, arxiv 2024.
>
> [3] Understanding Pre-training and Fine-tuning from Loss Landscape Perspectives, arxiv 2025.

---

### Decision · Program_Chairs · 2025-09-17

**Decision:**

Accept (poster)

**Comment:**

(a) Summary: This paper proposes RoMa, a robust watermarking scheme for diffusion models that protects intellectual property against fine-tuning attacks. Unlike existing watermarking methods that fail due to sharp local minima, RoMa decomposes watermarking into two key components: Embedding Functionality for reliable watermark detection and Path-specific Smoothness to ensure robustness along the watermark-connected path. Leveraging linear mode connectivity, RoMa establishes a flat and stable parameter region that preserves watermarks under fine-tuning while maintaining generation quality. Experiments demonstrate that RoMa significantly outperforms prior approaches in watermark robustness and verification efficiency.

(b) Strengths:
1. The study of watermarking problem about defending against fine-tuning attacks is important. RoMa achieves good performance against this attack.

2. The idea of using mode connectivity to analyze the impact of model finetuning on watermarks is interesting and likely valid.

(c) Weaknesses (after rebuttal):
1. Reviewers have suggested that some of the motivations for this paper lack sufficient validation (at least in the writing).

(d) Why this decision:
The paper is on the borderline. All reviewers express that their major concerns have been addressed in the rebuttal and some of reviewers are willing to increase their scores. Given that all reviewers are positive and the strengths outweight the weaknesses, AC would recommend Accept (poster).

(e) Summary of discussions:
See above